



# A Statistical Comparison of Cirrus Particle Size Distributions Measured Using the 2D Stereo Probe During the TC⁴, SPartICus, and MACPEx Flight Campaigns with Historical Cirrus Datasets

M. Christian Schwartz[1,2]

Computation Institute, University of Chicago, 5735 South Ellis Avenue, Chicago, IL, 60637, United States
Argonne National Laboratory, 9700 Cass Avenue, Bldg. 240, 6.A.15, Lemont, IL, 60439, United States

*Correspondence to*: M. Christian Schwartz (mcs45@byu.net)





**Abstract.** This paper addresses two straightforward questions. First, how similar are the statistics of cirrus particle
size distribution (PSD) datasets collected using the 2D Stereo (2D-S) probe to cirrus PSD datasets collected using
older Particle Measuring Systems (PMS) 2D Cloud (2DC) and 2D Precipitation (2DP) probes? Second, how similar
are the datasets when shatter-correcting post-processing is applied to the 2DC datasets? To answer these questions,
a database of measured and parameterized cirrus PSDs, constructed from measurements taken during the Small
Particles in Cirrus (SPartICus), Mid-latitude Airborne Cirrus Properties Experiment (MACPEx), and Tropical
Composition, Cloud, and Climate Coupling (TC$^4$) flight campaigns is used.

18          Bulk cloud quantities are computed from the 2D-S database in three ways: first, directly from the 2D-S

data; second, by applying the 2D-S data to ice PSD parameterizations developed using sets of cirrus measurements
collected using the older PMS probes; and third, by applying the 2D-S data to a similar parameterization developed
using the 2D-S data itself. Thereby a parameterized version of what the 2DC would have seen had it flown on the
above missions next to the 2D-S is compared to a similarly parameterized version of the 2D-S. It is seen, given the
same cloud field and given the same assumptions concerning ice crystal cross-sectional area, density, and radar
cross section, that the parameterized 2D-S and the parameterized 2DC predict similar distributions of inferred
shortwave extinction coefficient, ice water content, and 94 GHz radar reflectivity. However, the parameterization of
the 2DC based on uncorrected data predicts a statistically significant higher number of total ice crystals and a larger
ratio of small ice crystals to large ice crystals than does the parameterized 2D-S. The 2DC parameterization based
on shatter-corrected data also predicts statistically different numbers of ice crystals than does the parameterized 2D-
S, but the comparison between the two is nevertheless more favorable. It is concluded that the older data sets
continue to be useful for scientific purposes, with certain caveats, and that continuing field investigations of cirrus
with more modern probes is desirable.





## 1 Introduction

For decades, in situ ice cloud particle measurements often indicated ubiquitous, high concentrations of the

smallest ice particles (Korolev et al., 2013a; Korolev and Field, 2015). If the smallest ice particles are indeed
always present in such large numbers, then their effects on cloud microphysical and radiative properties are
pronounced. Heymsfield et al. (2002) reported small particles' dominating total particle concentrations ($N_T$s) at all
times during multiple Tropical Rainfall Measuring Mission (TRMM) field campaigns, while Field (2000) noted the
same phenomenon in mid-latitude cirrus. Lawson et al. (2006) reported $N_T$s in mid-latitude cirrus ranging from ~
.2-1 cm$^{-3}$ and showed that particles smaller than 50 microns were responsible for 99% of $N_T$, 69% of shortwave
extinction, and 40% of ice water content (IWC). From several representative cirrus cases, Gayet et al. (2002)
reported average $N_T$s as high as 10 cm$^{-3}$ and estimated that particles having maximum dimensions smaller than 15.8
microns resulted in about 38% of measured shortwave extinction; and Gayet et al. (2004) and Gayet et al. (2006)
estimated from a broader set of measurements that particles smaller than 20 microns accounted for about 35% of
observed shortwave extinction.

However, shattering of ice particles on probe tips and inlets and on aircraft wings has rendered many

historical cirrus datasets suspect (Vidaurre and Hallet, 2009; Korolev et al., 2011) due to such shattering's
artificially inflating measurements of small ice particle concentrations (see, e.g., McFarquhar et al., 2007; Jensen et
al., 2009; and Zhao et al., 2011). Measured ice particle size distributions (PSDs) are used to formulate
parameterizations of cloud processes in climate and weather models, so the question of the impact of crystal
shattering on the historical record of ice PSD measurements is one of significance (Korolov and Field, 2015).

Post-processing of optical probe data based on measured particle inter-arrival times (Cooper, 1978; Field et

al., 2003; Field et al., 2006; Lawson, 2011; Jackson et al., 2014; Korolev and Field, 2015) has become a tool for
ameliorating contamination from shattered artifacts. Shattered particle removal is based on modeling particle inter-
arrival times by a Poisson process, assuming that each inter-arrival time is independent of all other inter-arrival
times. Jackson and McFarquhar (2014) posit that particle clustering (Hobbs and Rangno, 1985; Kostinski and Shaw,
2001; Pinsky and Khain, 2003; Khain et al., 2007), which would violate this basic assumption, is not likely a matter
of significant concern as cirrus particles are naturally spread further apart than liquid droplets and sediment over a
continuum of size-dependent speeds.

A posteriori shattered particle removal should be augmented with design measures such as specialized





probe arms and tips (Vidaurre and Hallet, 2009; Korolov et al., 2011; Korolev et al., 2013a; Korolev and Field,
2015).  Probes must also be placed away from leading wing edges (Vidauure and Hallet, 2009; Jensen et al., 2009),
as many small particles generated by shattering on aircraft parts are likely not be filtered out by shatter-recognition
algorithms.

The ideal way to study the impact of both shattered particle removal and improved probe design is to fly

two versions of a probe—one with modified design and one without—side by side and then to compare results from
both versions of the probe both with and without shattered particle removal.  Results from several flight legs made
during three field campaigns where this was done are described in three recent papers:  Korolev et al. (2013b),
Jackson and McFarquhar (2014), and Jackson et al. (2014).  Probes built for several particle size ranges were
examined, but those of interest here are cloud particle probes: the 2D-S and the older Two-Dimensional Cloud
(2DC) probe.  Three particular results distilled from those papers are useful here.

First, a posteriori shattered particle removal is more effective at reducing counts of apparent shattering

fragments for the 2D-S than are modified probe tips, which result jibes with Lawson (2011).  The opposite is true for
the 2DC.  This is attributed to the 2D-S' larger sample volume and improvements in resolution and time response
over the 2DC (Jensen et al., 2009; Lawson, 2011), which allow it to size particles smaller than 100 μm and to
measure particle inter-arrival times more accurately (Lawson et al., 2010; Korolev et al., 2013b).

Second, shattered artifacts seem mainly to corrupt particle size bins less than about 500 microns.  Thus

Korolev et al. (2013b) posit that bulk quantities computed from higher order PSD moments, such as shortwave
extinction coefficient, IWC, and radar reflectivity, are likely to compare much better between the 2D-S and the 2DC
than is $N_T$.

Third, the efficacy of shattered particle removal from the 2DC is questionable:  post-processing is prone to

accepting shattered particles and to rejecting real particles (Korolev and Field, 2015).  The parameters of the
underlying Poisson model and its ability to correctly identify shattered fragments depend on the physics of the cloud
being sampled (Vidaurre and Hallett, 2009; Korolev et al., 2011).  Issues with instrument depth-of-field, unfocused
images, and image digitization further compound uncertainty (Korolev et al., 2013b; Korolev and Field, 2015).

In the context of relatively small studies such as these, Korolev et al. (2013b) pose two questions:  "(i) to

what extent can the historical data be used for microphysical characterization of ice clouds, and (ii) can the historical
data be reanalyzed to filter out the data affected by shattering?"  One difficulty in addressing these questions is the



scarcity of data from side-by-side instrument comparisons. Another is that, especially for the 2DC, "correcting
[data] a posteriori is not a satisfactory solution" (Vidaurre and Hallet, 2009). However, shattered particle removal is
the main (if not the only) correction method available when revisiting historical datasets.
As a start, though, Korolev et al.'s (2013b) first question is here addressed in terms of bulk cloud
properties, using shatter-corrected 2D-S data. Two points are critical to recall. First, the 2D-S is reasonably
expected to give results superior to the 2DC after shattered particle removal. Second, lingering uncertainty
notwithstanding, results presented elsewhere from the shatter-corrected 2D-S reveal behaviors in ice microphysics
within different regions of cloud that are expected both from physical reasoning and from modeling studies and that
were not always discernible before from in situ datasets (Lawson, 2011; Schwartz et al., 2017a).
To this end, a substantial climatology of shatter-corrected, 2D-S-measured cirrus PSDs is indirectly
compared with two large collections of older datasets, collected from the early 1990s through the mid-2000s mainly
using Particle Measurement Systems 2DC and Two-Dimensional Precipitation (2DP) probes (Baumgardner, 1989)
as well as Droplet Measurement Technologies Cloud- and Precipitation-Imaging Probes (CIP and PIP; Heymsfield
et al., 2009), and in one instance, the 2D-S. The older datasets are presented and parameterized in Delanoë et al.
(2005; hereinafter D05) and in Delanoë et al. (2014; hereinafter D14). The data used in D05 were not subject to
shattered particle removal, whereas the data in D14 were a posteriori.
The D05/D14 parameterizations take PSD moments as inputs and output parameterized 2DC PSDs. So, to
make the comparison, requisite moments from the 2D-S data are applied to the D05/D14 models in order to simulate
what the shatter- and non-shatter-corrected 2DCs would have measured had they flown with the 2D-S. A similarly
parameterized version of what the 2D-S actually measured is computed in order to make a fair comparison. It is
then seen whether the older datasets differ statistically from the newer via derived cirrus bulk properties.
Section 2 contains a description of the data used herein. Section 3 discusses the fitting of PSDs with
gamma distributions for computational use, Section 4 discusses the normalization and parameterization schemes
used by D05/D14, and Section 5 discusses the effects of not having included precipitation probe data with the 2D-S
data. Section 6 demonstrates the final results of the comparison.
**2 Data**
The 2D-S data was collected during the Mid-Latitude Airborne Cirrus Experiment (MACPEx), based in
Houston, TX during February and March, 2011 (MACPEx Science Team, 2011); the Small Particles in Cirrus




(SPartICus) campaign, based in Oklahoma during January through June, 2010 (SPartICus Science Team, 2010); and
TC[4], based in Costa Rica during July, 2007 (TC[4] Science Team, 2007). The SPEC 2D-S probe (Lawson, 2011)
images ice crystal cross-sections via two orthogonal lasers that illuminate two corresponding linear arrays of 128
photodiodes. PSDs, as well as distributions of cross-sectional area and estimated mass, are reported every second in
size bins with centers starting at 10 microns and extending out to 1280 microns. Particles up to about three
millimeters can be sized in one dimension by recording the maximum size along the direction of flight. During
SPartICus the 2D-S flew aboard the SPEC Inc. Learjet, while during MACPEx it was mounted on the NASA WB57
aircraft. During TC[4] it was mounted on both the NASA DC8 and the NASA WB57, but the WB57 data is not used
due to documented contamination of the data from shattering artifacts off of the aircraft wing (Jensen et al., 2009).
Temperature was measured during MACPEx, TC[4], and SPartICus using a Rosemount total temperature
probe. Bulk IWC measurements are available for MACPEx from the Closed-path tunable diode Laser Hygrometer
(CLH) probe (Davis et al., 2007). Condensed water that enters the CLH is evaporated so that a measurement of total
water can be made. The condensed part of the total water measured by the CLH is obtained by estimating
condensed water mass from concurrent PSDs measured by the National Center for Atmospheric Research (NCAR)
Video Image Particle Sampler (VIPS) probe and then subtracting this estimate from the measured total water mass.
**3 Parametric Fitting of PSDs**
PSDs measured by the 2D-S were fit with both unimodal and bimodal parametric gamma distributions.
The unimodal distribution is
$$n\left(D\right) = N_0 \left(D/D_0\right)^{\alpha} \exp\left(-D/D_0\right), \quad (1)$$

where $D$ is particle maximum dimension, $D_0$ is the scale parameter, $\alpha$ is the shape parameter, and $N_0$ is the so-called
intercept parameter. The bimodal distribution is simply a mixture of two unimodal distributions:
$$n\left(D\right) = N_{01} \left(D/D_{01}\right)^{\alpha_1} \exp\left(-D/D_{01}\right) + N_{02} \left(D/D_{02}\right)^{\alpha_2} \exp\left(-D/D_{02}\right). \quad (2)$$

Save in a handful of instances (which will be indicated), all bulk PSD quantities shown here are computed using
these parametric fits. A combination of unimodal and bimodal fits is used to compute $N_T$, dictated by the shape of
the PSD as determined by a generalized chi-squared goodness of fit test (Schwartz et al., 2017b). Unimodal fits are
used to compute all other bulk quantities.



Unimodal fits were performed via the method of moments [in a manner similar to Heymsfield et al.
(2002)].  Both the method of moments and an expectation maximization algorithm (Moon, 1996; see the Appendix)
were used for the bimodal fits; the more accurate of those two fits [as determined by whether fit provided the
smaller binned Anderson-Darling test statistic (Demortier, 1995)] being kept.
Measured PSDs are both truncated and time-averaged in order to mitigate counting uncertainties.  It is here
assumed that temporal averaging sufficiently reduces Poisson counting noise so that it may be ignored [see, e.g.,
Gayet et al. (2002)].  Given already cited concerns regarding uncertainty in shattered particle removal, the smallest
size bins are not automatically assumed here to be reliable.  Other competing uncertainties further complicate
particle counts within the first few size bins, e.g., the possible underestimation of counts of real particles by a factor
of 5-10 (Gurganus and Lawson, 2016) and mis-sizing of larger drops into smaller size bins due to image break-up at
the edge of the instrument's depth of field (Korolev et al., 2013b; Korolev and Field, 2015).
In order to determine how many of the smallest size bins to truncate and for how many seconds to average
in order to make the counting assumption valid, two simple exercises were performed using the MACPEx dataset.
In the first exercise, fifteen-second temporal averages were performed along with truncating zero through two of the
smallest size bins while only the unimodal fits (chosen according to a maximum likelihood ratio test [Wilks, 2006])
were kept.  Figure 1 shows comparisons of distributions of measured and computed (from the fits) $N_T$s.  The
difference in the number of samples of computed $N_T$ between zero bins and one bin truncated is an order of
magnitude higher than that between one bin and two bins truncated.  This is due to frequent, extraordinarily high
numbers of particles recorded in the smallest size bin that at times cause a PSD to be flagged as bimodal by the
maximum likelihood ratio test.  As this effect lessens greatly after truncating only one bin, and as the computed and
measured $N_T$s are otherwise better matched using a single-bin truncation, the smallest size bin is ignored for all
PSDs (making the smallest size bin used 15-25 microns).
Also, IWC was estimated from the fit distributions (the first size bin having been left off in the fits) using
the mass-dimensional relationship $m\left(D\right) = 0.0065 D^{2.25}$ ($m$ denotes mass, and all units are cgs) given in
Heymsfield (2003) for mid-latitude cirrus.  The distribution of IWC thus computed nominally matches (not shown)
IWC estimates from both the CLH and from the 2D-S data product, which uses mass-projected area relationships
(Baker and Lawson, 2006).
For the second exercise, temporal averages from one to 20 seconds were performed, truncating the first size



bin and again keeping only the unimodal fits. The balance to strike in picking a temporal average length is
acceptably to smooth out Poisson counting uncertainties without losing physical information to an overlong average.
Qualitatively, the statistics of the fit parameters begin to steady at around 15 seconds (not shown), so a fifteen-
second temporal average was chosen. Using the data filters, temporal average, and bin truncation thus far described
results in ~17 000 measured PSDs and their accompanying fits.

It must be noted that the first 2D-S size bin contains at least some real particles, though the afore-

mentioned uncertainties make it impossible (at present) to know how many. Therefore, $N_{TS}$ computed from the
remaining bins are perforce underestimations. Parametric fits extrapolate the binned data all the way to size zero,
though; so it could be assumed, if the real ice particle populations are in fact gamma-distributed, that this
extrapolation is a fair estimate of the real particles lost due to truncating the first size bin. In truth, however, the
assumption of a gamma-shaped PSD is arbitrary, if convenient; and even the assumption of Poisson counting
statistics, in the face of the artifacts mentioned in this paper, may be unwarranted. It is therefore felt that the
averaging approach is justified. The gamma PSD shape is kept for its convenience and for its ability to reproduce
higher-order PSD moments. However, in this paper, where $N_{TS}$ (equivalently, the zeroth moments) from either the
parametric, the binned, or the normalized parametric PDSs are computed, the computations are begun at the left
edge of the second size bin so as to compare equivalent quantities. Thus, total number concentrations presented for
comparison here are truncated to compensate for having left off the smallest size bin.
**4  Normalization and Parameterization**

Each 2D-S-measured PSD $n_D(D)$, whose independent variable is ice particle maximum dimension, is

transformed to a distribution $n_{D_e}(D_e)$ whose independent variable is equivalent melted diameter. The
transformations are performed twice: once using the density-dimensional relationship used in D05 and once using a
mass-dimensional relationship used in D14.

The density-dimensional relationship $\rho(D) = aD^b$ ($\rho$ denotes density, $D$ denotes particle maximum

dimension, $a = 0.0056$, $b = -1.1$, and all units are cgs) used in D05 stems from relationships published by
Locatelli and Hobbs (1974) and Brown and Francis (1995) for aggregate particles. Setting masses equal as in D05
results in the independent variable transformation





$$D_e = \left( \frac{aD^b}{\rho_w} \right)^{1/3} D , \quad (3)$$

where $\rho_w$ is the density of water.
The mass-dimensional relationship labeled "Composite" (Heymsfield et al., 2010) in D14 is used here for
the second transformation:
$$m(D) = 7e^{-3}D^{2.2} .$$

(All units are again cgs.) Setting masses equal results in the independent variable transformation
$$D_e = \left( \frac{6a_{mi}}{\pi \rho_w} \right)^{1/3} D^{b_{mi}/3} . \quad (4)$$

The "composite" relation was only used to normalize about 54% of the PSD's utilized in D14; however, those
datasets so normalized are broadly similar to MACPex, SPartICus, and TC[4] (one in fact is TC[4], where the Cloud
Imaging Probe was used as well as the 2D-S), and so the "composite" relation is used here for comparison with D14.
Following D05/D14s' notation, transformed PSDs then have their independent variable scaled by mass-
mean diameter
$$D_m = \frac{\int_0^\infty D_e^4 n_{D_e}(D_e) dD_e}{\int_0^\infty D_e^3 n_{D_e}(D_e) dD_e} \quad (5)$$

and their ordinates scaled by
$$N_0^* = \frac{4^4}{\Gamma(4)} \frac{\left[ \int_0^\infty D_e^3 n_{D_e}(D_e) dD_e \right]^5}{\left[ \int_0^\infty D_e^4 n_{D_e}(D_e) dD_e \right]^4} , \quad (6)$$

so that
$$n_{D_e}(D_e) = N_0^* F\left( x = \frac{D_e}{D_m} \right) . \quad (7)$$

In Eq. (7), $F(x)$ is, ideally, the universal, normalized PSD (Meakin, 1992; Westbrook et al., 2004a,b; D05; Tinel
et al, 2005; D14).



Having binned PSDs, the normalization procedure is wended as described in section 4.1 of D05.  First, the
2D-S bin centers and bin widths are transformed once using Eq. (3) for the comparison with D05 and once again
using Eq. (4) for the comparison with D14.  Next, each binned PSD is transformed by scaling from $D$-space to $D_e$-
space (see below).  Then, via numerically computed moments, Eqs. (5)-(7) are used to normalize the binned, mass-
equivalent spherical PSDs, which are then grouped into normalized diameter bins of $\Delta x_i = 0.10$.
The scale factor for transformation is derived based on this simple consideration:  if the number of particles
within a size bin is conserved upon the bin's transformation from $D$-space to $D_e$-space, then, given that the
transformation is from maximum dimension to mass-equivalent spheres, so also is the mass of the particles within a
size bin conserved.  That is,

$$n_{D_e}\left(D_{e_i}\right) = n_D\left(D_i\right)\frac{aD_i^{b+3}\Delta D_i}{\rho_w D_{e_i}^3 \Delta D_{e_i}} \quad (8)$$

for the D05 transformation and

$$n\left(D_{e_i}\right) = n\left(D_i\right)\frac{a_m D^{b_m}\Delta D_i}{\left(\frac{\pi}{6}\right)\rho_w D_{e_i}^3 \Delta D_{e_i}} \quad (9)$$

for the D14 transformation.
Mass-equivalent transformations theoretically ensure that both $N_T$ and IWC can be obtained by using the
PSD in either form:

$$N_T = \int_0^\infty n_D\left(D\right)dD = \int_0^\infty n_{D_e}\left(D_e\right)dD_e \quad (10)$$

$$\text{IWC} = \frac{\pi}{6}\int_0^\infty aD^{b+3}n_D\left(D\right)dD = \frac{\pi}{6}\int_0^\infty \rho_w D_e^3 n_{D_e}\left(D_e\right)dD_e \quad . \quad (11)$$

As it turns out, scaling from $D$-space to $D_e$-space so that Eqs. (10) and (11) are both satisfied is not necessarily
possible.  Since for the sake of estimating $D_m$ and $N_0^*$ it is more important that the IWCs be matched, this was done
for the D05 comparison while matching the $N_{TS}$ to within a factor of approximately 0.75, plus a bias of ~3.1 L$^{-1}$.
The following transformation of variables must be used for computing other bulk quantities from
transformed PSDs (Bain and Englehardt, 1992):





$$n_D(D) = n_{D_e}[D_e(D)]\left|\frac{dD_e}{dD}\right| . \quad (12)$$

For instance, effective radar reflectivity is computed using a set of power-law fits of T-matrix computations of
backscatter cross section to particle maximum dimension (Matrosov, 2007; Matrosov et al., 2012; Posselt and Mace,
2013; Hammonds et al., 2014) as follows:
$$Z_e = \frac{10^8 \lambda^4}{|K_w|^2 \pi^5} \sum_i \int_{D_i}^{D_{i+1}} a_{zi} D^{b_{zi}} n_{D_e}[D_e(D)]\left|\frac{dD_e}{dD}\right| dD.$$

The coefficients $(a_{zi}, b_{zi})$ were derived assuming an air/ice dielectric mixing model and that all particles are prolate
spheroids with aspect ratios of 0.7 (Korolev and Isaac, 2003; Westbrook et al., 2004a; Westbrook et al., 2004b;
Hogan et al., 2012). Several explicit expressions for computing bulk quantities based on equivalent distributions
may be found in Schwartz (2014).
In D05/D14, data taken with cloud particle and precipitation probes were combined to give PSDs ranging
from 25 μm to several millimeters. No precipitation probe data is used here, but how does not including
precipitation probe data affect the comparison? This question will be addressed later in this paper.
Two-dimensional histograms of the normalized PSD are shown in Fig. 2 for the D05 transformation and in
Fig. 4 for the D14 transformation, overlaid with their mean normalized PSDs (cf. Figs. 1 and 2 in D05 and Fig. 3 in
D14). For both transformations, the mean normalized PSDs for the three datasets combined are repeated in Figs. 3
and 5 as solid curves (cf. Fig. 3 of D05 and Fig. 6 of D14). These serve as the empirical universal, normalized PSDs
$F_{\sim 2DS-D05}(x)$ and $F_{\sim 2DS-D14}(x)$, derived using the mass transformations of D05 and D14, respectively. They,
and the quantities derived therefrom, serve to represent the more modern 2D-S with shattered particle removal. The
subscripts $\sim$2DS-D05 and $\sim$2DS-D14 are used hereinafter to represent bulk quantities derived using $F_{\sim 2DS-D05}(x)$
and $F_{\sim 2DS-D14}(x)$.
Three parametric functions for $F(x)$ are given in D05, two of which are repeated here: the gamma-$\mu$
function ($F_\mu$) and the modified gamma function ($F_{\alpha,\beta}$; Petty and Huang, 2011).



$$F_\mu(x) = \frac{\Gamma(4)}{4} \frac{(4+\mu)^{4+\mu}}{\Gamma(4+\mu)} x^\mu \exp\left[-(4+\mu)x\right] \quad (13)$$

$$F_{\alpha,\beta}(x) = \beta \frac{\Gamma(4)}{4^4} \frac{\Gamma\left(\frac{\alpha+5}{\beta}\right)^{4+\alpha}}{\Gamma\left(\frac{\alpha+4}{\beta}\right)^{5+\alpha}} x^\alpha \exp\left[-\left(\frac{\Gamma\left(\frac{\alpha+5}{\beta}\right)}{\Gamma\left(\frac{\alpha+4}{\beta}\right)}\right)^\beta\right] \quad (14)$$

Values of $\mu$, $\alpha$, and $\beta$ can be chosen to fit these functions to a mean normalized PSD. In D05, the parametric
functions $F_{\alpha,\beta} = F_{(-1,3)}$ (Eq. (14)) and $F_\mu = F_3$ (Eq. (13)) are given to approximate the universal PSD derived
from combined 2DC-2DP datasets; and in D14, the parametric function $F_{\alpha,\beta} = F_{(-0.262,1.754)}$ is given to
approximate the universal PSD derived from shatter-corrected datasets collected mainly with combined 2DC-2DP
probes.
These functions are used to parameterize transformed PSDs measured by the 2DC-2DP, given two PSD
moments. We therefore make the assumption that if we take the same two moments derived from a 2D-S-measured
PSD and then apply them to Eq. (13) or (14), we have effectively simulated the transformed PSD that a combined
2DC-2DP would have observed had they been present with the 2D-S. The subscripts ~2DC(u) and ~2DC(s) are
used hereinafter to represent quantities that simulate 2DC-2DP data (non-shatter-corrected and shatter-corrected,
respectively) in this way. Initial observations on comparison of $F_{\sim2DS-D05}(x)$ and $F_{\sim2DS-D14}(x)$ with the
normalizations of D05 and D14 will now be given.
**4.1 Comparison with D05**
Right off the bat, some important qualitative observations can be made from examining $F_{\sim2DS-D05}(x)$ in
Fig. 3. First, in contrast to Fig. 3 of D05, the concentrations of particles at the smallest scaled diameters of
$F_{\sim2DS-D05}(x)$ are, on average, about an order of magnitude or more lower. From this it is surmised that while the
2D-S continues to register relatively high numbers of small ice particles, the number has decreased in the newer
datasets due to the exclusion of shattered ice crystals.
It can also be seen in Fig. 3 that the shoulder in the normalized PSDs in the vicinity of $x \sim 1.0$ exists in the



newer data as it does in the data used in D05.  It is worth noting, though, that the shoulder exists in the one tropical
dataset used here (TC[4]), whereas it is absent or much less noticeable in the tropical datasets used in D05.
Fortuitously, $F_{\alpha,\beta} = F_{(-1,3)}$ fits the 2D-S data better than it does the older data at the smallest normalized
sizes (cf. Fig. 2 in D05).  Neither $F_{\alpha,\beta} = F_{(-1,3)}$ nor $F_{\mu} = F_3$ correctly catches the shoulder in the newer data,
though $F_{\alpha,\beta} = F_{(-1,3)}$ was formulated to (better) catch a corresponding shoulder in the older data.
Next, a comparison of PSD quantities computed directly from the 2D-S with corresponding ~2DC-derived
quantities (computed using $N_0^*$ and $D_m$ derived directly from the binned 2D-S data and applied to $F_{\alpha,\beta} = F_{(-1,3)}$
and $F_{\mu} = F_3$) is made.  The extinction coefficient, IWC, and 94 GHz radar reflectivity compare well between the
2D-S and both versions of ~*2DCu* (not shown).  2D-S and ~*2DCu* radar reflectivities have a slightly skewed and
slightly non-one-to-one relationship (this is an important consideration in the parameterization of $N_0^*$ by $Z$ given in
D05).  As for $N_T$, it is the least certain computation (see Fig. 6); but $F_{\mu} = F_3$ is entirely wrong in attempting to
reproduce this quantity, so this shape is not used hereinafter and $F_{\sim 2DCu}(x) = F_{(-1,3)}(x)$ is the shape used to
simulate the uncorrected 2DC-2DP.
Figure 7 shows the mean relative error and the standard deviation of the relative error (cf. Fig. 5 of D05)
between 2D-S-derived and corresponding ~*2DCu*-derived quantities.  (Effective radius is as defined in D05, and 2D-
S estimates of $N_T$ here stem from truncated, binned data).  Mean relative error for both extinction coefficient and
IWC is about -0.1%.  The mean relative error in $N_T$ ($N_T$ computed directly from binned PSDs is used both here and
in Fig. 8) is rather large at ~50%; and the mean relative error in $Z_e$, at ~22%, is larger than that shown in Fig. 5 of
D05 (less than 5% there) but, at about 2 dB, is within the error of most radars.  This may well be due to the
overestimation of $F(x)$ by $F_{\sim 2DCu}(x)$ between normalized sizes of about 1.2 and 2 [see Fig. 3b].  Both here and
in D05, $F_{\sim 2DCu}(x)$ falls off much more rapidly than $F(x)$ above a normalized diameter of two.  However, it is
deduced from Figs. 2 and 5 in D05 that this roll-off is not responsible for the large mean relative error in $Z$ shown in
Fig. 7.
The mean relative error in effective radius shown in Fig. 7 is approximately -7%, whereas it is apparently


nil in Fig. 5 of D05. Effective radius is defined in D05 as the ratio of the third to the second moments of the
spherical-equivalent PSDs and is therefore a weighted mean of the PSD. The negative sign on the relative error
indicates that, on average, $F_{\sim 2DCu}(x)$ is underestimating the effective radius of the PSDs measured by the 2D-S
whereas for the older datasets it hits the effective radius spot-on (in the average). Therefore, there is a significant
difference between the 2D-S datasets and the older 2DC-2DP datasets in the ratio of large particles to small
particles, even when precipitation probe data is not combined with the 2D-S.
**4.2 Comparison with D14**
From Fig. 4, concentrations at the smallest scaled diameters of $F_{\sim 2DS-D14}(x)$ are nominally consistent
with those shown in Fig. 6 of D14. In accordance with the surmise made in the comparison with D05 above, it
would seem that shattered particle removal from the 2DC improves comparison between the 2D-S and the 2DC-2DP
at the smallest particle sizes.
Here, $F_{\sim 2DCs}(x) = F_{(-0.262, 1.754)}(x)$. The shoulder in the normalized PSDs in the vicinity of $x \sim 1.0$ is
again found, though the shoulder is not captured by $F_{\sim 2DCs}(x)$ (see Fig. 5). The normalized 2D-S at the smallest
normalized sizes is also underestimated by $F_{\sim 2DCs}(x)$. Comparison of $N_T$ computed using $F_{\sim 2DCs}(x)$ with that
derived from 2D-S is quite similar to that of $F_{\sim 2DCu}(x)$ (not shown).
As shown in Fig. 8, the mean relative error between $N_T$ and effective radius derived from the 2D-S and
from $\sim 2DCs$ is again about 50%, while the mean relative error in effective radius remains about -7.5%. The mean
relative error in reflectivity has decreased to about 14%.
**5 Impact of Not Using Precipitation Probe Data**
To more formally investigate the impact of not using a precipitation probe, data from the PIP were
combined with data from the 2D-S using the TC[4] dataset. This campaign of the three was chosen due to its tending
to occur at warmer temperatures, in a more convective environment, and at lower relative humidities: so, if large
particles are going to matter, they should matter for TC[4]. Figure 9 shows, similar to Figs. 3 and 5, $F_{\sim 2DS-D05}(x)$
for the 2D-S alone, $F_{\sim 2DS-D05}(x)$ for the 2D-S combined with the PIP, and $F_{\sim 2DCu}(x)$.
In the combined data, the average, normalized PSD does not dig as low between zero and unity as for the





2D-S alone; but it does show similar numbers of particles at the very smallest normalized sizes, and the shoulder is
in the same location.  Beginning at about x = 1.2, the 2D-S-PIP normalized distribution is higher than the 2D-S-
alone normalized distribution; and it continues out to about x = 10, whereas the 2D-S-alone distribution ends shy of
x = 5.  In either case, $F_{\sim 2DCu}(x)$ misses what is greater than about x = 2.  This roll-off, along with the fact that the
mean normalized and transformed 2D-S/PIP combination appears to be more similar to $F_{\sim 2DS-D05}(x)$ than it does
to $F_{\sim 2DCu}(x)$, indicate that a parameterization of $F(x)$ based off the 2D-S alone is comparable to the 2DC/2DP-
based $F_{\sim 2DCu}(x)$ parameterization.

In support of this assertion, Fig. 10 shows the penalty in radar reflectivity, computed directly from data

using the approach described earlier, incurred by using only the 2D-S instead of the 2D-S-PIP.  The penalty is in the
neighborhood of 1 dB.

The true $N_0^*$ and $D_m$ computed from each of the 2D-S PSDs alone and from the combined PSDs from

TC[4] were used, along with $F_{\sim 2DCu}(x)$, to compute $N_T$, extinction coefficient, IWC, and 94 GHz effective radar
reflectivity.  This amounts to two different ~2DCu simulations:  one including the PIP and one not.  The results are
shown in Fig. 11.  The distributions are very similar, with the exception of the reflectivity distributions, whose
means are separated by less than 1 dBZ.  It is concluded that the cloud filtering technique has resulted in PSDs that
are satisfactorily described by the 2D-S alone, at least in the case of this comparison.
**6  Final Results and Discussion**

In D05, complete parameterization of a 2DC-2DP-measured PSD is achieved by using a universal shape

$F_{\alpha,\beta}(x)$ along with $N_0^*$ parameterized by radar reflectivity and $D_m$ parameterized by temperature.  For
comparison with the shattered-corrected D14 study, a temperature-based parameterization of "composite"-derived
$D_m$ is also derived from the 2D-S data and "composite"-derived $N_0^*$ is parameterized by radar reflectivity.  A
similar parameterization scheme (also based on radar reflectivity and temperature) for the 2D-S (based on Field et
al., 2005) is outlined in Schwartz et al. (2017c) and is used here to compute a fully parameterized version of 2D-S-
measured PSDs so as to make a fair comparison of them with fully parameterized 2DC-measured PSDs.

Figure 12 shows the results of computing PSD-based quantities using the fully parameterized 2D-S (red,





labeled "*x2DS*"), the fully parameterized (uncorrected) 2DC (blue, labeled "*x2DCu*"), and directly from the 2D-S
data (black).   Probability density functions (pdfs) of 94 GHz effective radar reflectivity match because they are
forced to by the two instrument parameterizations.  Otherwise, biases exist between the two sets of computations
based on simulated instruments and computations based on the actual 2D-S (black curve).  This bias is due mainly to
the temperature parameterization of $D_m$.  The pdfs of extinction coefficient and IWC for the two parameterized
instruments match one another quite well (the differences in their medians are not statistically significant).
However, for $N_T$, the *x2DCu* pdf is shifted to higher concentrations than the pdf for *x2DS*.  The difference in their
medians is statistically significant at the 95% level according to a Mann-Whitney U test.  It is therefore concluded
that the older D05 parameterization based on the 2DC-2DP data sets predicts a statistically significant higher
number of total ice crystals than does the parameterized 2D-S (by a factor of about 1.3, or a little over 1 dB) and
that, more generally, the 2DC measures a larger ratio of small ice crystals to large ice crystals than does the 2D-S, as
shown in the effective radius comparison in Fig. 7.

Figure 13 shows pdfs of $N_T$ and extinction coefficient computed using the fully parameterized 2D-S (red,

labeled "*x2DS*"), the fully parameterized (corrected) 2DC (blue, labeled "*x2DCs*"), and directly from the 2D-S data
(black).  The pdfs of extinction match quite well, but their medians are significantly different according to the U test.
The medians of NT are also significantly different, but the mean of the parameterized, corrected 2DC is lower than
that of the parameterized 2D-S.  A posteriori shatter correction has made 2DC measurements more like 2D-S
measurements in the bulk quantity of total particle concentration, however, a statistically significant difference
between the 2D-S and the corrected 2DC remains.  This result is entirely expected in light of the previous results
outlined in the introduction to this paper.

Via an indirect comparison to older, 2DC-based datasets by means of parameterizations given in D05 and

in D14, it is determined that the 2D-S cirrus cloud datasets used here are significantly different from historical
datasets in numbers of small ice crystals measured.  Furthermore, it is determined that were a 2DC to have been
flown alongside a 2D-S during MACPEx, SPartICus, and TC[4], the 2DC would have reported significantly higher
numbers of the smallest ice crystals.  Were a posteriori shattered particle removal applied to the 2DC data the total
numbers of ice crystals measured by the 2D-S and 2DC would have become more similar, but $N_T$ measured by the
2DC would remain statistically different from that measured by the 2D-S.

Our aim was to determine whether the historical data sets analyzed by D05 and D14 continue to be





scientifically viable given the newer probes and modern processing techniques. Given the modest differences found
here between the newer and older data, we conclude that the historical data sets do indeed continue to be useful with
the caveats noted above. However, it is surmised that, since the efficacy of a posteriori shatter correction on the
2DC is questionable and since the 2D-S is superior in response time, resolution, and sample volume to the 2DC, and
that since steps were taken to mitigate ice particle shattering in the 2D-S data, the newer datasets are more accurate.
Therefore, continuing large-scale field investigations of cirrus clouds using the newer particle probes and data
processing techniques are recommended and, where possible, investigation, by means of flying 2DC probes
alongside 2D-S probes, of the possibility of effecting statistical correction of historical cirrus ice particle datasets
using newer datasets.

It is important to note that this study does not specifically consider PSD shape. This is a critical component

of the answers to Korolov et al.'s (2013b) original two questions. Mitchell et al. (2011) demonstrated that for a
given effective diameter and IWC, the optical properties of a PSD are sensitive to its shape. Therefore, PSD
bimodality and concentrations of small ice crystals are critical to realistically parameterizing, cirrus PSDs, to
modeling their radiative properties and sedimentation velocities, and to mathematical forward models designed to
infer cirrus PSDs from remote sensing observations (Lawson et al., 2010; Mitchell et al, 2011; Lawson, 2011). We
therefore reiterate the need for ongoing, large-scale investigations of cirrus clouds that make use of advanced
imaging equipment, such as the 2D-S, flown alongside older instruments such as the 2DC. Thus, not only will new
measurements with up-to-date instruments be made, but the measurements necessary for the correction of historical
cirrus datasets will also be obtained.
**Data Availability**

All SPartICus data may be accessed via the Atmospheric Radiation Measurement (ARM) data archive as

noted in the references. All MACPEx and TC$^4$ data may be accessed from the NASA Earth Science Project Office
(ESPO) data archive, also noted in the references.
**ACKNOWLEDGEMENTS**

Thanks are given to Drs. Gerald G. Mace, Paul Lawson, and Andrew Heymsfield for helpful discussions

that led to significant improvement in the manuscript. This work was supported by the U.S. Department of Energy's
(DOE) Atmospheric System Research (ASR), an Office of Science, Office of Biological and Environmental
Research (BER) program, under Contract DE-AC02-06CH11357 awarded to Argonne National Laboratory. This





work was also supported by the National Science Foundation (NSF) Grant AGS-1445831. Grateful
acknowledgement is also given to the computing resources provided on Blues, a high-performance computing
cluster operated by the Laboratory Computing Resource Center (LCRC) at the Argonne National Laboratory.
**APPENDIX: Bimodal Fits using Expectation Maximization**

At the beginning, it is noted that this algorithm is effective at identifying and parameterizing multiple

modes in binned ice PSD measurements made by a single instrument, but that it does not work well for PSDs that
result from combining binned measurements of two different probes (as of the 2D-S and a precipitation probe.)
Following Johnson et al. (2013), binned counts of cloud particles are modeled as independent samples taken from a
multinomial distribution (Bain and Englehardt, 1992). Let $y_l$ be the number of particle counts in the $l^{th}$ size bin
(from a total of $L$ possible size bins). Thus, if the observed PSD is the vector $\mathbf{y}$, then its probability mass function
(pmf) is
$$\Pr(\mathbf{y}) = p_{\mathbf{y}}(\mathbf{y}) = \frac{N_T!}{\prod_{l=1}^{L} y_l!} \prod_{l=1}^{L} p_l^{y_l}, \quad (A1)$$

where $p_l$ is the probability of obtaining a count in the $l^{th}$ size bin (computed by integrating Eq. (3)) and where $N_T$ is
the total number of counts.

Alternatively, binned counts can be modeled as samples from a multinomial distribution with $2L$ bins: a

sample may fall in bin $l_1$ (small particle mode) or in bin $l_2$ (large particle mode). Let $x_{1l}$ be the number of counts in
the $l^{th}$ small-mode size bin, and let $x_{2l}$ be the number of counts in the $l^{th}$ large-mode size bin. In this case, the pmf
may be expressed in either of two forms:
$$p_{\mathbf{x}_1,\mathbf{x}_2}(\mathbf{x}_1,\mathbf{x}_2) = \frac{N_T!}{\prod_{l=1}^{L} x_{l1}! x_{l2}!} \prod_{l=1}^{L} p_{l1}^{x_{l1}} p_{l2}^{x_{l2}}, \quad \text{or} \quad (A2)$$

$$p_{\mathbf{x}_1,\mathbf{x}_2}(\mathbf{x}_1,\mathbf{x}_2) = \frac{N_T!}{\prod_{l=1}^{L} x_{l1}! x_{l2}!} \exp\left\{ \sum_{l=1}^{L} x_{l1} \ln[p_{l1}] + x_{l2} \ln[p_{l2}] \right\}. \quad (A3)$$

Here, $p_{l1}$ is the probability of obtaining a count in the $l^{th}$ size bin of the PSD's small mode and likewise for $p_{l2}$.
These probabilities are computed by integrating the large and small modes of Eq. (4). For example,
$$p_{l1} = \frac{\eta_1}{\Gamma(\alpha_1+1)} \left[ \gamma\left(\alpha_1+1, \frac{D_{l+1}}{D_1}\right) - \gamma\left(\alpha_1+1, \frac{D_l}{D_1}\right) \right], \quad (A4)$$



where $\gamma\left(...\right)$ is the lower incomplete gamma function.

This algorithm iterates through pairs of expectation and maximization steps. Following Moon (1996), we

begin with the function
$$Q\left(\boldsymbol{\theta},\boldsymbol{\theta}^{[k]}\right)=\mathrm{E}\left\{\ln\left[\,p\left(\mathbf{x}\mid\boldsymbol{\theta}\right)\mid\mathbf{y},\boldsymbol{\theta}^{[k]}\right]\right\}, \quad \text{(A5)}$$

where $p\left(\mathbf{x},\boldsymbol{\theta}\right)$ denotes the likelihood function of the vector of mixture distribution parameters $\boldsymbol{\theta}$ given the
(missing) data vector $\mathbf{x}=\left[\mathbf{x_1}\ \mathbf{x_2}\right]$. For the expectation step, the expected value of the log-likelihood function is
computed with respect to $\mathbf{x}$, given a set of observed data $\mathbf{y}$ and a current estimate of the parameters $\boldsymbol{\theta}^{[k]}$. This
expected value becomes the current estimate of the missing data $\mathbf{x}^{[k]}$. Equation (A5) is then maximized with respect
to the parameter vector $\boldsymbol{\theta}$ in order to obtain a new estimate $\boldsymbol{\theta}^{[k+1]}$:
$$\boldsymbol{\theta}^{[k+1]}=\arg\max_{\boldsymbol{\theta}}\left[Q\left(\boldsymbol{\theta},\boldsymbol{\theta}^{[k]}\right)\right]. \quad \text{(A6)}$$

As $p_{\mathbf{x}_1,\mathbf{x}_2}\left(\mathbf{x}_1,\mathbf{x}_2\right)$ belongs to the exponential distribution family, it suffices for the expectation step

simply to estimate $\mathbf{x}_1$ and $\mathbf{x}_2$ (Moon, 1996). To do so, their expected values, conditioned upon the observations $\mathbf{y}$
and on an estimate of the distribution parameters $\boldsymbol{\theta}^{[k]}$, are found. Equation (A6) is accordingly transformed so that
$$p_{\mathbf{x}_1,\mathbf{y}}\left(\mathbf{x}_1,\mathbf{y}\right)=p_{\mathbf{x}_1,\mathbf{x}_2}\left(\mathbf{x}_1,\mathbf{y}-\mathbf{x}_1\right)=\frac{N_T!}{\prod_{l=1}^{L}x_{l1}!\left(y_l-x_{l1}\right)!}\prod_{l=1}^{L}p_{l1}^{x_{l1}}p_{l2}^{\left(y_l-x_{l1}\right)}. \quad \text{(A7)}$$


The conditional distribution of $\mathbf{x}_1$ is therefore
$$p\left(\mathbf{x}_1\mid\mathbf{y}\right)=\frac{p_{\mathbf{x}_1,\mathbf{y}}\left(\mathbf{x}_1,\mathbf{y}\right)}{p_{\mathbf{y}}\left(\mathbf{y}\right)}=\prod_{l=1}^{L}\frac{y_l!}{x_{l1}!\left(y_l-x_{l1}\right)!}\left(\frac{p_{l1}}{p_l}\right)^{x_{l1}}\left(\frac{p_{l2}}{p_l}\right)^{\left(y_l-x_{l1}\right)}, \quad \text{(A8)}$$

which is the joint distribution of a sample of independent binomial random variables. Similarly, it may be found
that
$$p\left(\mathbf{x}_2\mid\mathbf{y}\right)=\prod_{l=1}^{L}\frac{y_l!}{x_{l2}!\left(y_l-x_{l2}\right)!}\left(\frac{p_{l1}}{p_l}\right)^{\left(y_l-x_{l2}\right)}\left(\frac{p_{l2}}{p_l}\right)^{x_{l2}}. \quad \text{(A9)}$$

The conditional expectations are therefore





$$E\left[x_{l1} \mid y_l\right] = x_{l1}^{[k]} = y_l \frac{p_{l1}^{[k]}}{p_l^{[k]}} \text{ , and } \quad (A10)$$

$$E\left[x_{l2} \mid y_l\right] = x_{l2}^{[k]} = y_l \frac{p_{l2}^{[k]}}{p_l^{[k]}} . \quad (A11)$$

The maximization step then requires maximizing the logarithm of Eq. (A3) with respect to $\boldsymbol{\theta}$, using the
estimates $\mathbf{x}_1^{[k]}$ and $\mathbf{x}_2^{[k]}$. Dropping terms that are not functions of the distribution parameters gives
$$\boldsymbol{\theta}^{[k+1]} = \arg\max_{\boldsymbol{\theta}} \left\{ \sum_{l=1}^{L} x_{l1}^{[k+1]} \ln\left[p_{l1}\right] + x_{l2}^{[k+1]} \ln\left[p_{l2}\right] \right\}. \quad (A12)$$

Equations (A10)-(A12) are iterated, using an initial estimate for the distribution parameters $\boldsymbol{\theta}^{[0]}$ (see Schwartz,
2014), until the distribution parameters converge.
Solutions for $N_1$ and $N_2$ come from the definition of $\eta_1$ and from the second non-central sample moment of
the non-normalized PSD $M_2$:
$$N_1 = \frac{\eta_1 M_2}{D_1 \Gamma\left(\alpha_1 + 1\right)\left[\eta_1 D_1^2\left(\alpha_1 + 2\right)\left(\alpha_1 + 1\right) + \left(1 - \eta_1\right)\left(\alpha_2 + 2\right)\left(\alpha_2 + 1\right)\right]} \quad (A14)$$

$$N_2 = \frac{\left(1 - \eta_1\right) M_2}{D_2 \Gamma\left(\alpha_2 + 1\right)\left[\eta_1 D_1^2\left(\alpha_1 + 2\right)\left(\alpha_1 + 1\right) + \left(1 - \eta_1\right)\left(\alpha_2 + 2\right)\left(\alpha_2 + 1\right)\right]} . \quad (A15)$$

The second non-central moment is used to ensure good reproduction of the zeroth through the second non-central
PSD moments by the solution. The mode for which the product of the scale and shape parameters is the larger is
selected as the large mode.
The log-likelihood function in Eq. (A5) has many local maxima so the iterative search for a global
maximum is highly sensitive to the first guess given to it. The method for dealing with this problem is described in
Schwartz (2014).
**Competing Interests**
The author declares that he has no conflict of interest.



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



**FIGURE CAPTIONS**

Figure 1: Comparisons of computed and measured total number concentration for 15-second PSD averages and for truncation of none through the first two PSD size bins.

Figure 2: Histograms of normalized PSDs from each flight campaign, overlaid with their mean, normalized PSDs (D05 normalization). The color map is truncated at 75% of the highest number of samples in a bin so as to increase contrast. (a) TC[4]  (b) MACPEx  (c) SPartICus  (d) all data combined

Figure 3: The mean, normalized PSD (D05 normalization) from all three datasets combined, overlaid with two parameterizations from D05: the gamma-mu parameterization (dash-dotted curve) and the modified gamma parameterization (dashed curve). Panel (b) is a zoom-in on a portion of panel (a).

Figure 4: Same as Figure 2, but using D14 normalization.

Figure 5: The mean, normalized PSD (D14 normalization) from all three datasets combined, overlaid with the parameterizations from D14. Panel (b) is a zoom-in on a portion of panel (a).

Figure 6: Total number concentration computed using the parameterized universal PSDs from D05 along with true values of $N_0^*$ and $D_m$ (from the 2D-S data) scattered vs. total number concentration computed directly from untransformed 2D-S data.

Figure 7: Mean relative error and standard deviation of the relative error between total number concentration (divided by 10), effective radius, IWC, and Z as computed directly from the 2D-S and as computed from the modified-gamma universal PSD shape and the true $N_0^*$ and $D_m$ computed from the 2D-S data.

Figure 8: As in Figure 7, but using the shatter-corrected 2DC parameterization.

Figure 9: Data from TC[4] alone. The mean, normalized PSD from the 2D-S is overlaid with the mean, normalized PSD obtained from combining the 2D-S with the PIP and the modified gamma parameterization from D05 (dashed curve). Panel (b) is a zoom-in on a portion of panel (a).

Figure 10: Two-dimensional histogram of 94 GHz effective radar reflectivity computed, using the Hammonds/Matrosov approach, from the 2D-S alone versus that computed from the 2D-S combined with the PIP.

Figure 11: Distributions of quantities computed using the parametric modified gamma distribution along with the true values of $N_0^*$ and $D_m$ computed from the 2D-S alone and from the 2D-S combined with the PIP. (a) $N_T$ (b) extinction coefficient  (c) IWC  (d) 94 GHz effective radar reflectivity

Figure 12: Marginal pdfs of quantities computed directly from 2D-S data, as well as computed using the parameterized 2D-S and the parameterized, uncorrected 2DC. (a) total number concentration (b) shortwave extinction coefficient  (c) ice water content  (d) radar reflectivity

Figure 13: Marginal pdfs of quantities computed directly from 2D-S data, as well as computed using the parameterized 2D-S and the parameterized, corrected 2DC. (a) total number concentration (b) shortwave extinction





**FIGURES**

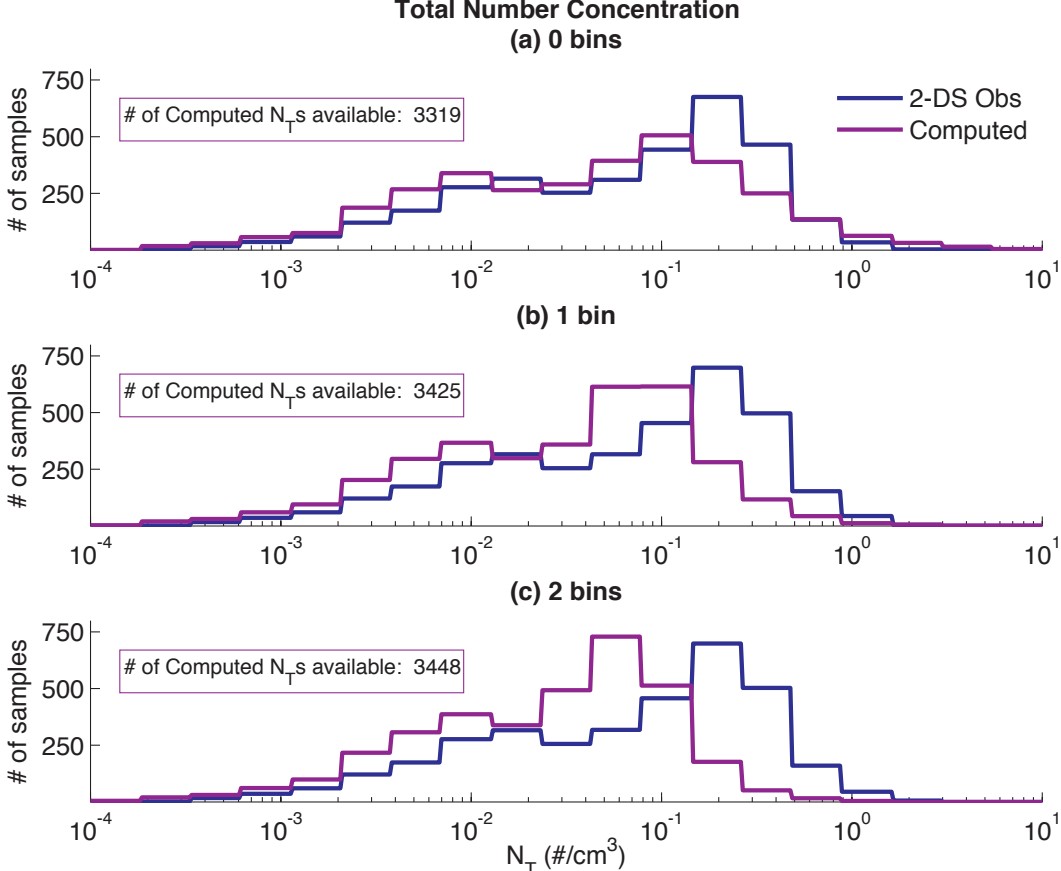

**Figure 1:  Comparisons of computed and measured total number concentration for 15-second PSD averages**
**and for truncation of none through the first two PSD size bins.**



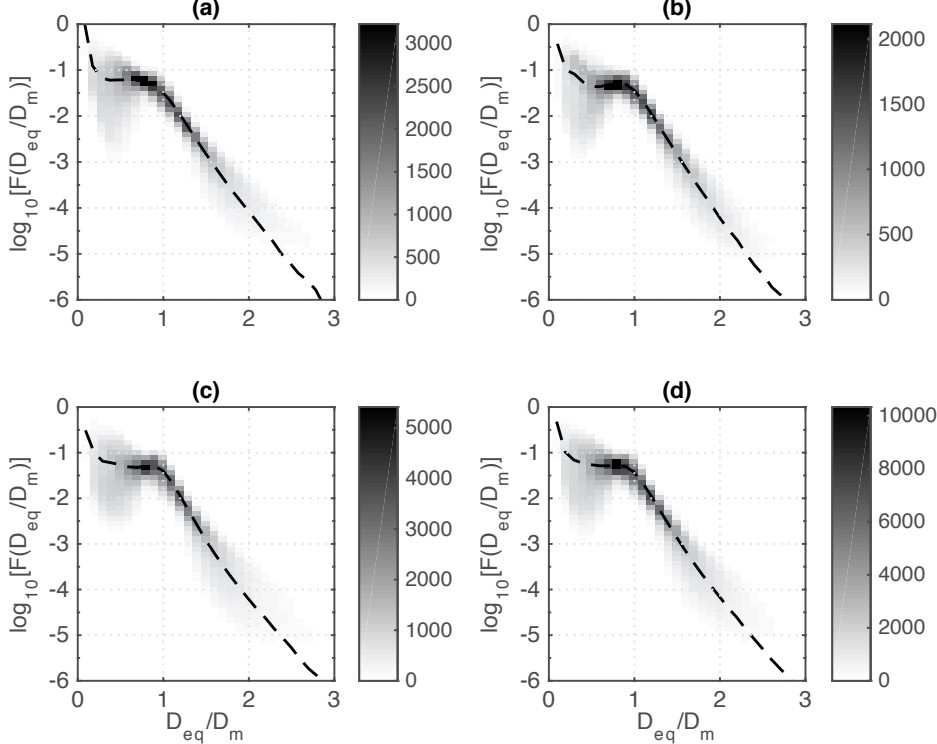

**Figure 2: Histograms of normalized PSDs from each flight campaign, overlaid with their mean, normalized PSDs (D05 normalization). The color map is truncated at 75% of the highest number of samples in a bin so as to increase contrast. (a) TC[4] (b) MACPEx (c) SPartICus (d) all data combined**





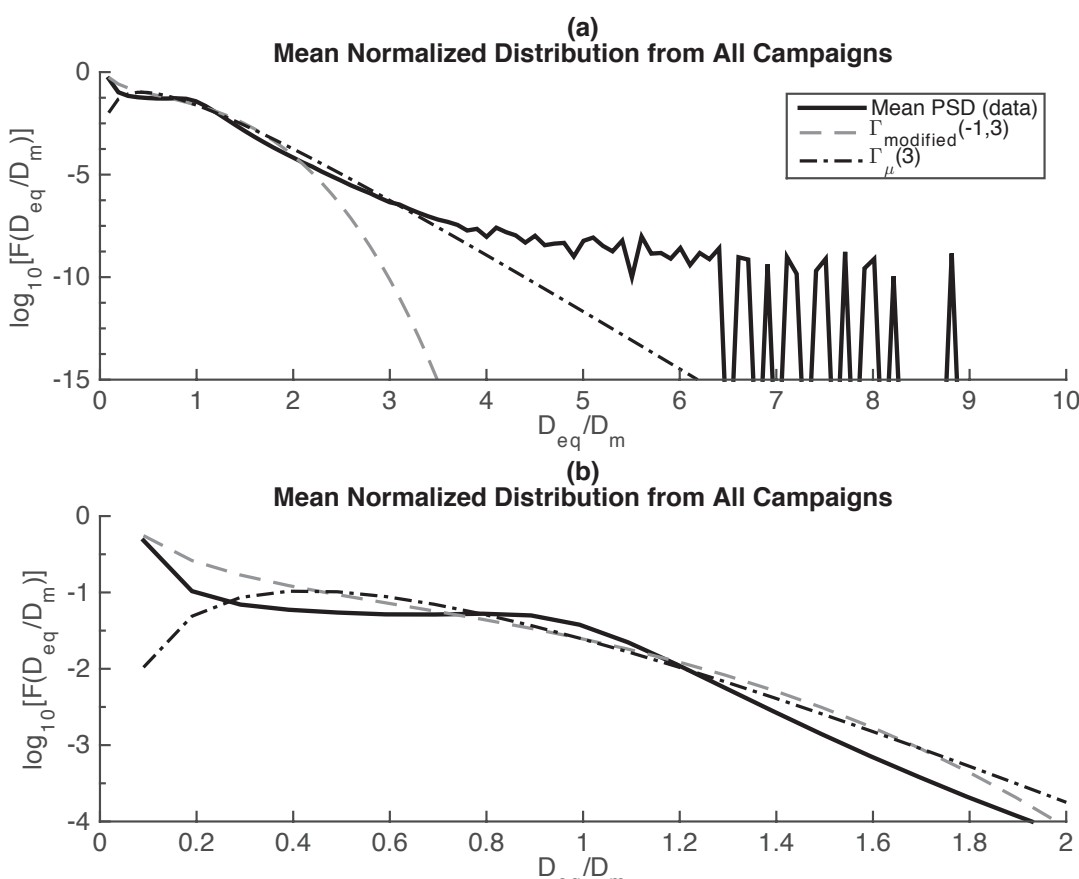

**Figure 3:** **The mean, normalized PSD (D05 normalization) from all three datasets combined, overlaid with two parameterizations from D05: the gamma-mu parameterization (dash-dotted curve) and the modified gamma parameterization (dashed curve). Panel (b) is a zoom-in on a portion of panel (a).**





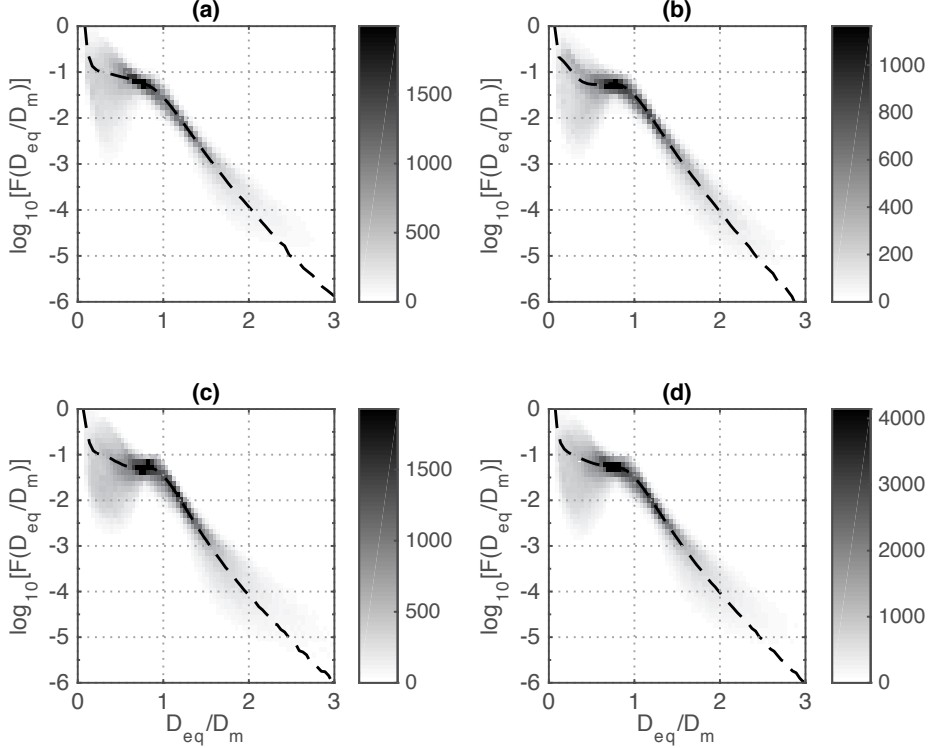

**Figure 4: Same as Figure 2, but using D14 normalization.**




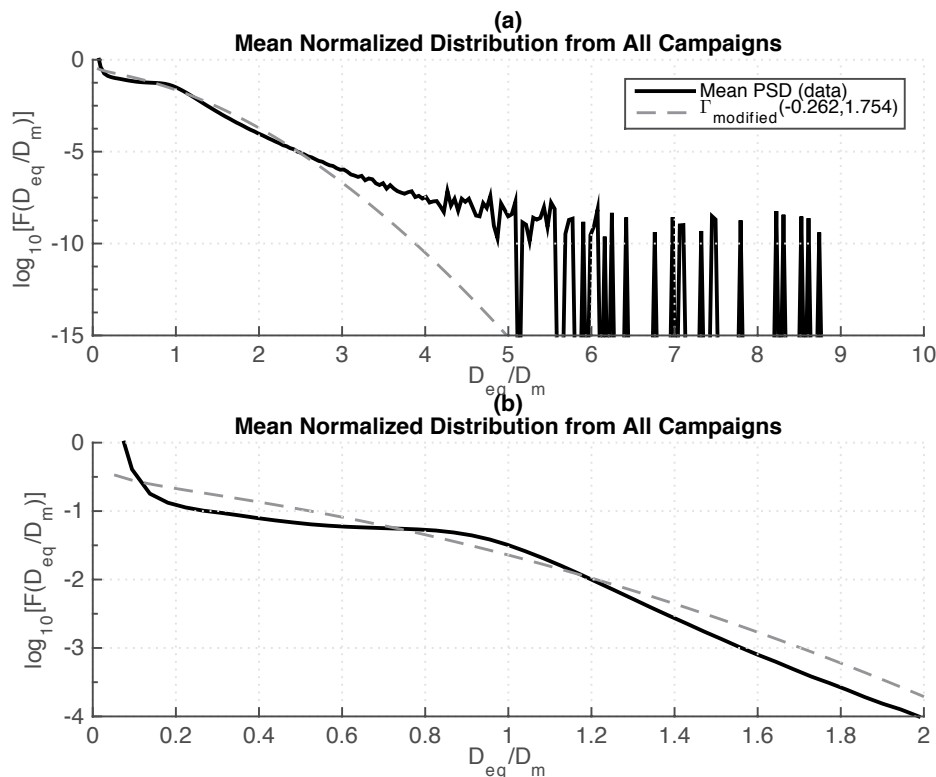

Figure 5: The mean, normalized PSD (D14 normalization) from all three datasets combined, overlaid with
the parameterizations from D14. Panel (b) is a zoom-in on a portion of panel (a).








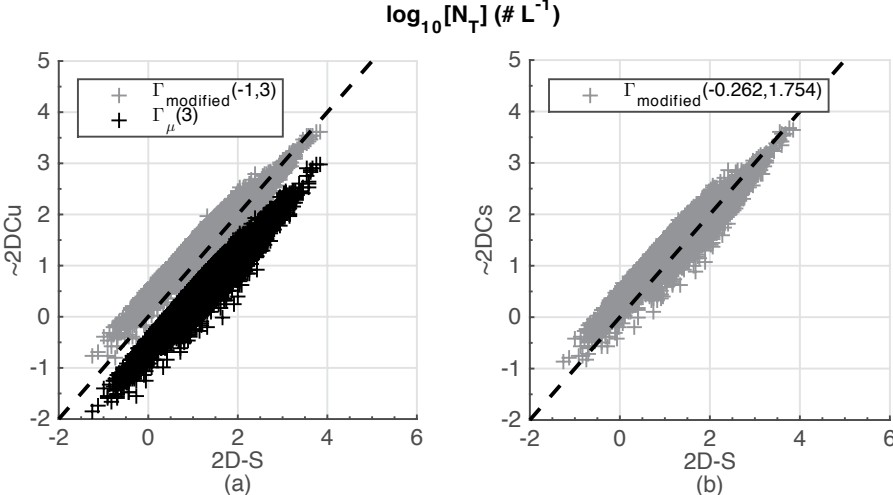

**Figure 6: Total number concentration computed using the parameterized universal PSDs from D05 along**
**with true values of $N_0^*$ and $D_m$ (from the 2D-S data) scattered vs. total number concentration computed**
**directly from untransformed 2D-S data.**





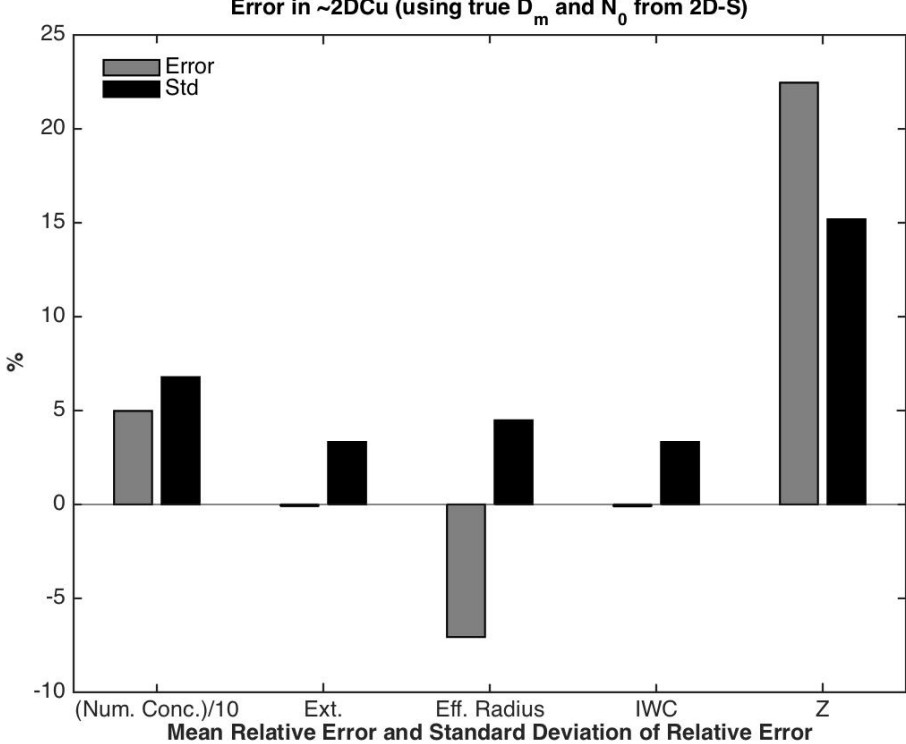

**Figure 7: Mean relative error and standard deviation of the relative error between total number**
**concentration (divided by 10), effective radius, IWC, and Z as computed directly from the 2D-S and as**
**computed from the modified-gamma universal PSD shape and the true $N_0^*$ and $D_m$ computed from the 2D-**
**S data.**






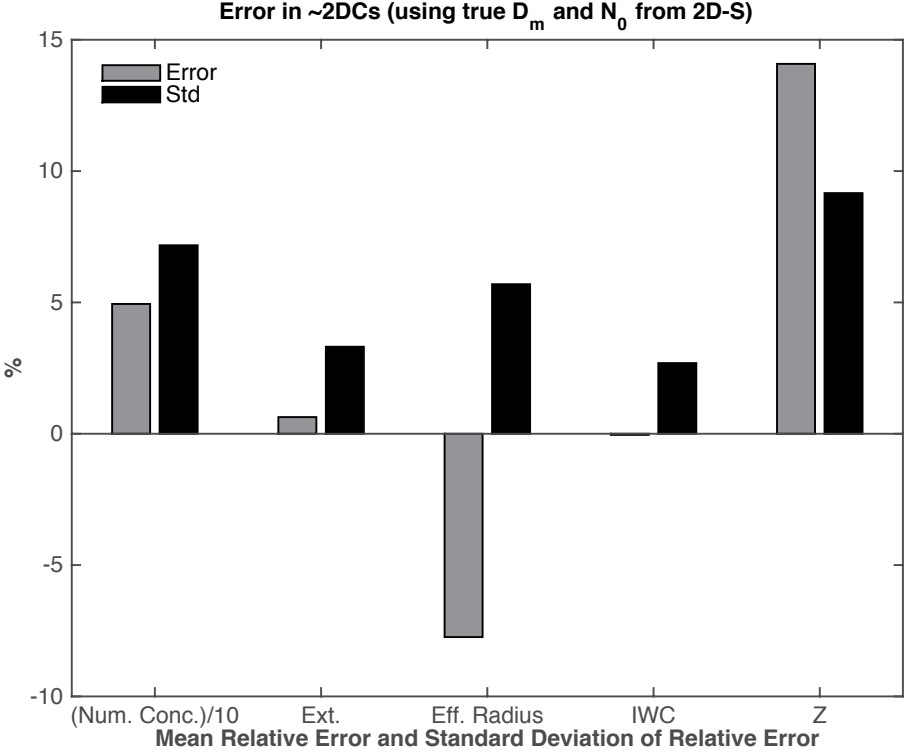

**Figure 8: As in Figure 7, but using the shatter-corrected 2DC parameterization.**




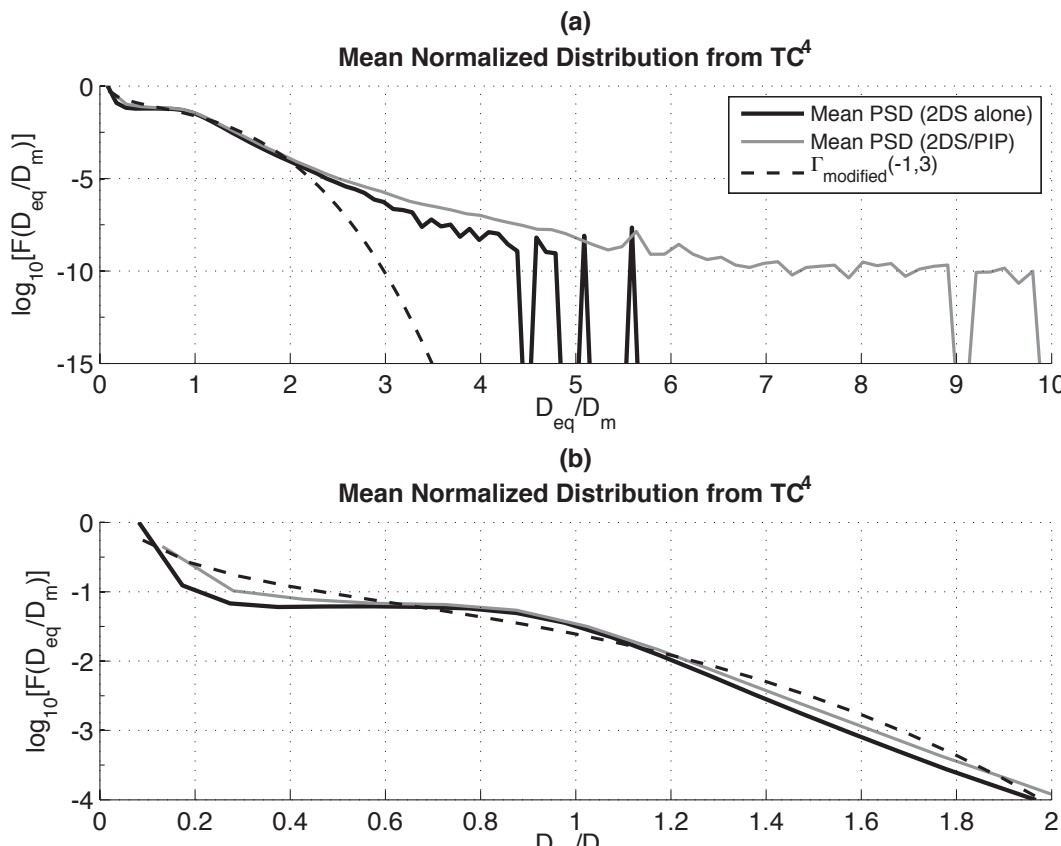

**Figure 9:  Data from TC[4] alone. The mean, normalized PSD from the 2D-S is overlaid with the mean,**
**normalized PSD obtained from combining the 2D-S with the PIP and the modified gamma parameterization**
**from D05 (dashed curve).  Panel (b) is a zoom-in on a portion of panel (a).**






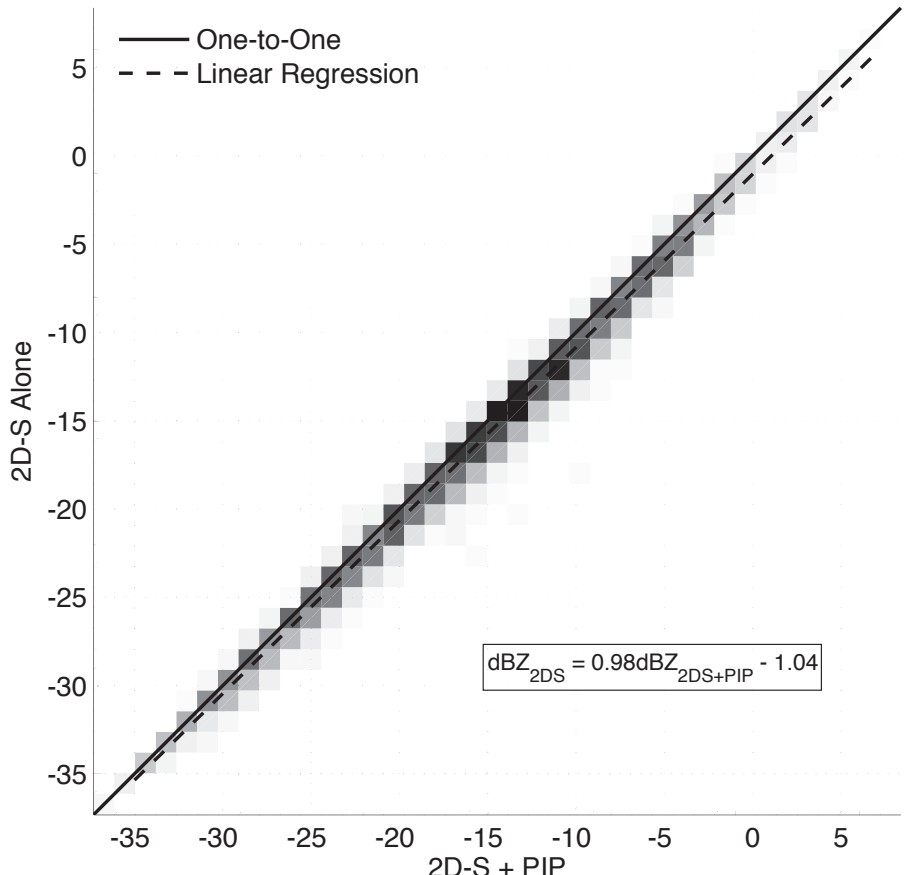


**Figure 10:  Two-dimensional histogram of 94 GHz effective radar reflectivity computed, using the Hammonds/Matrosov approach, from the 2D-S alone versus that computed from the 2D-S combined with the PIP.**





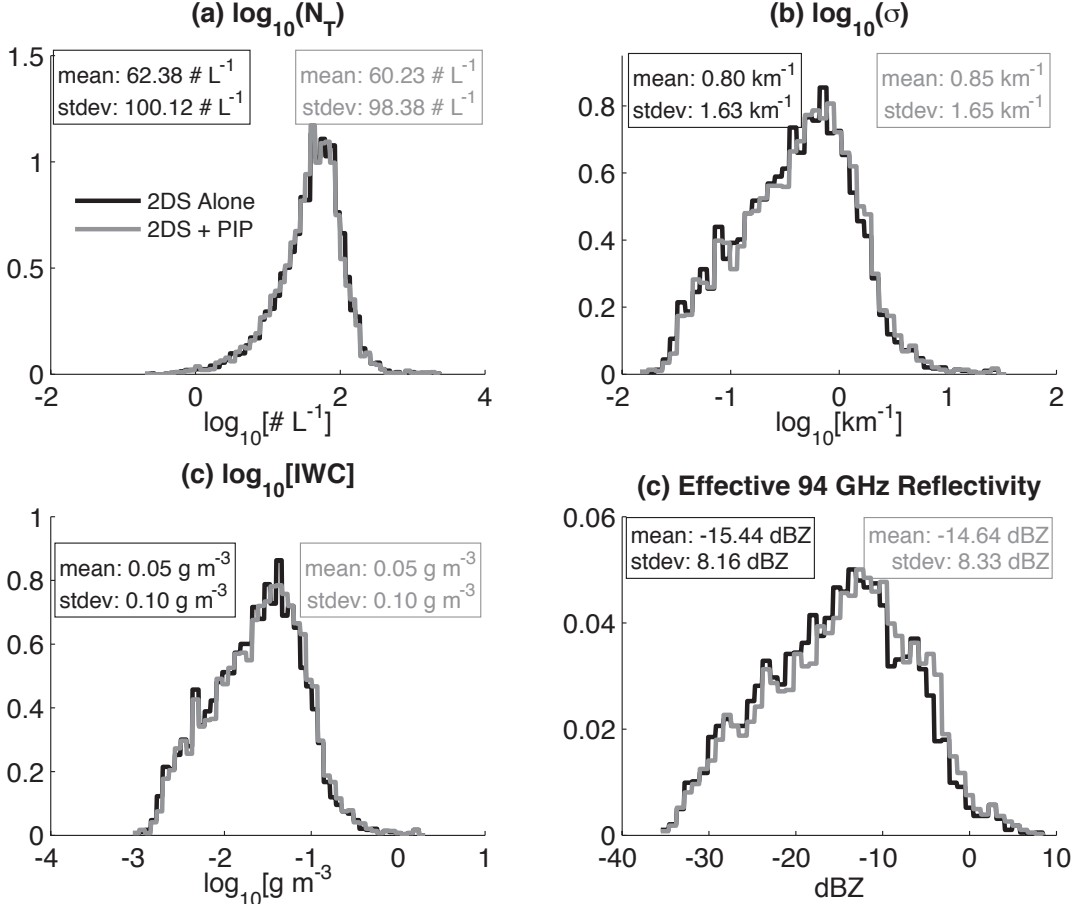

Figure 11: **Distributions of quantities computed using the parametric modified gamma distribution along with the true values of $N_0^*$ and $D_m$ computed from the 2D-S alone and from the 2D-S combined with the PIP. (a) $N_T$ (b) extinction coefficient (c) IWC (d) 94 GHz effective radar reflectivity**




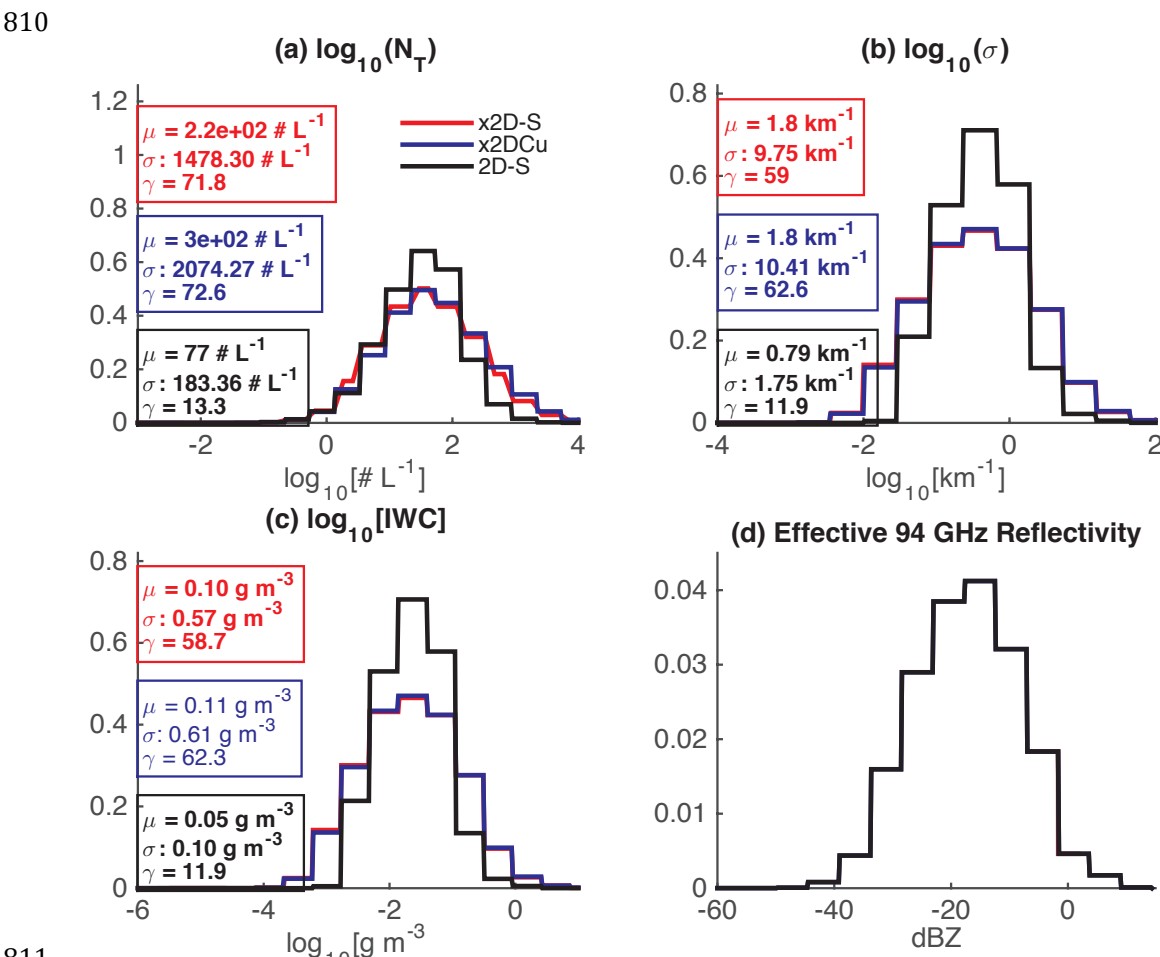

**Figure 12:  Marginal pdfs of quantities computed directly from 2D-S data, as well as computed using the**
**parameterized 2D-S and the parameterized 2DC.  (a) total number concentration (b) shortwave extinction**
**coefficient  (c) ice water content  (d) radar reflectivity**






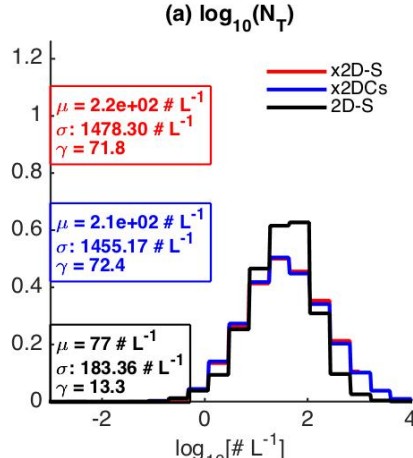
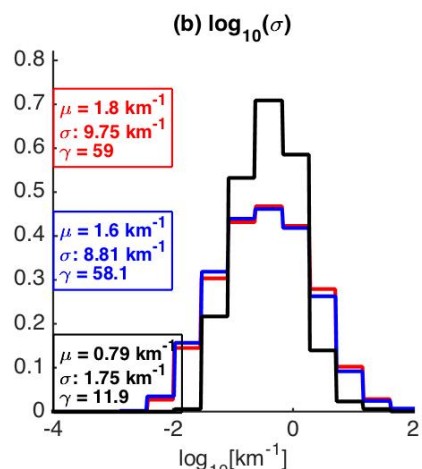

**Figure 13: Marginal pdfs of quantities computed directly from 2D-S data, as well as computed using the**
**parameterized 2D-S and the parameterized, corrected 2DC. (a) total number concentration (b) shortwave**
**extinction**