# Peer review of "A Statistical Comparison of Cirrus Particle Size Distributions 1"

_Atmospheric Measurement Techniques, 2017_

## Referee Comment (RC1) · Anonymous Referee #2 · 6 Apr 2017

The material in this manuscript is suitable for publication in amt. It gives a useful comparison between an older particle probe, the 2DC, and the newer probe, the 2DS, thought to provide more accurate ice crystal information. A compilation of the parameterization and normalization of many ice crystal size distributions measured by both probe types is used in an attempt to adjust the older probe data to make that data more reliable.

1. The paper needs a careful review concerning the lack of definition of some given

variables. For example, what is a_mi and b_mi in Eq. (4), what is D_eq in the Figures, what is subscript I ?

2. The accuracies of the density/dimension and mass/dimension relationships used in the paper are not discussed, even though they may affect the conclusions reached. A comment on such a possible affect.

3. The data for D05/D014 is listed as starting at 25 um; whereas the data for the 2DS starts at 15 um. Is this taken into account in the comparisons?

4. The author points out the difficulty of the probes measuring the smallest ice crystals, given that the probes can create errors due to uncorrected crystal shattering and other reasons. His sentence associated with small crystals (line 181) "It is therefore felt that the averaging approach is justified" is inconsistent with this difficulty.

5. The paper only deals with integrated ice-crystal properties, but it also points out that the nature of the ice-crystal size distribution should also play a significant role in probe performance. The latter is not dealt with in the paper. It would be helpful for the author to comment on what might be done to improve the size information on the smallest ice crystals that can dominate under certain atmospheric conditions (e.g., Heymsfield et al., 2010, JAS, 67, 3303-3318). For example, can forward scattering probes that respond to small particles be used for ice crystal measurements (e.g., Gerber and DeMott, 2014, JTECH, 31, 2145-2155) ?

6. The impressive Appendix is not essential for the conclusions reached in the paper. Deletion of the Appendix is recommended.

---

## Referee Comment (RC2) · D. Baumgardner (Referee) · 12 Apr 2017

This study is a logical and complementary follow-up of the Delanoë et al. (2005,2014) evaluations that provided parameterizations of cirrus size distributions based on a large set of measurements taken in both mid-latitude and tropical environments. The author has provided a detailed analysis using more recent measurements with a more modern imaging probe to address an important question:

"Given what we now know about the impact of crystal shattering on measurements by cloud particle spectrometers, can historical data sets be trusted"?

I think that this study has answered that question, at least with respect to cirrus clouds. In addition, even though the instrument that is used in this study has a faster response time than the earlier 2D-C and 2D-P, and marginally larger sample volume, the results of the current study would suggest that such instrument improvements really have minor impact on the overall statistical robustness of the previous measurements and may also only be marginally more accurate, especially given the many other uncertainties that the new instrument has not overcome. In particular, there remain major uncertainties due to unknown ice density and shape n the third dimension that lead to large error bars in derived bulk parameters.

It is only at the very smallest sizes where there is a clear difference between current and previous measurements; however, even when there are several orders of magnitude difference in concentration at these sizes, the propagated error in effective radius, IWC and reflectivity is surprisingly small.

What I think would be a useful, and perhaps even necessary, addition to this paper would be to include in Figs. 7&8 the relative errors and standard deviations that are reported in Delanoë et al. (2005,2014) where they compare their data sets against the parameterization. That would then put into context the current comparison with the parameterizations with the original, hence bringing closure.

The other very important source of uncertainty that the author side steps is that of over-sizing of out-of-focus ice crystals (Korolev, 2007). Although a correction for this issue has not yet been provided, such as has been done for water droplets, measurements in cirrus clearly show crystal images that are out of focus and that should be size-corrected. These might even be the source of the "bump" in the size distributions, i.e. a certain fraction of the particles in that size interval most certainly are smaller crystals out of focus. This bump is also seen in the Delanoë et al. (2005,2014) studies;

[Figure]

however, whereas the bump occurs in the current study at a Deq/DM <1, in Delanoë et al. (2005,2014) the bump is right at 1. How does the author explain this?

Lastly, the author refers to three of his papers that have not yet been published. These references should be removed since, as a reviewer, I was unable to access them.

———————————————

---

## Referee Comment (RC3) · Anonymous Referee #3 · 17 Apr 2017

General Comments:

Overall the paper is suitable for publication with minor changes. The microphysical probe comparisons presented are similar to past work, but the analyses are done in a slightly different and more systematic way. My main comment is that the paper would benefit from a more thorough introductory section, with historical insight into the probes discussed and the characteristics that make them different. This should include not only the ice shattering issue, but a brief summary of other technical differences.

[Figure]

Minor Comments:

Lines 21-22: Without reading the paper, this sentence in the Abstract is confusing and does not logically follow. Please clarify or simplify abstract.

Line 44: Add Garrett et al.: Small, highly reflective ice crystals in low-latitude cirrus, GRL 2003.

Line 72: "which results jibes" is awkward–please rephrase.

Line 104-108: Perhaps a simple diagram would be helpful here to eludicate the method and steps used?

Line 164: Why not use the actual size distributions?

Line 166: Please quantify "nominally matches", particularly since the data aren't shown.

Line 339: Is it really the "true" value?

Line 369: Missing subscript in NT.

Line 376: Delete this sentence as it's not really necessary?

Lines 387: A long and wordy sentence. Suggest breaking it up for clarity.

Line 391: If your other work giving better alternatives to the Gamma distribution is now published, please refer to it here.

Line 396-398: Redundant with statements in prior paragraph; remove.

Acknowledgements: No acknowledgements to those scientists who provided the field data?

---

## Referee Comment (RC4) · J. Reid (Referee) · 18 Apr 2017

Review: Minor revisions I agree with the previous reviewers in general. This is a pretty clean cut topic for this paper which adds statistical consistency to a long standing problem in ice measurement. Further, looking at the references, this is clearly one paper in a long series originating with the author's days at Utah and beyond. While the topic is clean cut, the paper is nevertheless difficult to follow at times, and the author could do much to improve readability, and hopefully in time, his h-index. Indeed, it is

overly terse at times. One previous reviewer noted that the introduction could use a bit of background. I certainly concur with that. Even though this has been reviewed in several other papers, it is good for a paper to be complete. Not only to be tied in more completely with the previous literature base, but also with the author's current lien of thought. I would also say the final results and discussion could also be worked on. For example, the author states to the effect that old data is still usable, provided previously described caveats are respected. Actually going through the paper several times, it was not clear what all of these are. Even though the smaller ice sizes can be mitigated for the bulk moments, what does this mean for say a forward optical model? Perhaps a separate conclusions, or discussion and conclusions as distinct from results, be provided that provides a bulleted list of what are the key take away points-sort of a recipe card. I would also suggest that figure captions be more verbose spelling out variables when convenient. Similarly, laying out in a bulletining form or table the different instruments and processing would help.

Other than these comments, my opinion matches those of the previous reviews: the paper oscillates between very formal writing, and conversational vernacular(e.g., jibes, right off the bat, etc); a diagram laying out the steps; . One point that requires emphasis as pointed out by reviewer 3 is the lack of data provider documentation in the acknowledgements. Indeed, by downloading data from the NASA servers not only did the author agree to acknowledge where the data came from, but actually offer coauthor ship to the data providers. Often for this sort of thing they will simply ask for acknowledgement, but the offer does need to be made.

Be well.

---

## Author Comment (AC1) · 1 Jun 2017

I thank the reviewer kindly for his helpful feedback. I will answer your comments in order in the attached supplement. The reviewers remarks appear in red, the responses in black. The plain text version may be found below

General Comments: Overall the paper is suitable for publication with minor changes. The microphysical probe comparisons presented are similar to past work, but the analyses are done in a slightly different and more systematic way. My main comment is that the paper would benefit from a more thorough introductory section, with historical insight into the probes discussed and the characteristics that make them different. This should include not only the ice shattering issue, but a brief summary of other technical differences.

Thank you! The following paragraphs have been added to the introduction.

[revised manuscript text omitted]

Minor Comments: Lines 21-22: Without reading the paper, this sentence in the Abstract is confusing and does not logically follow. Please clarify or simplify abstract.

Lines 21-22 have been changed to "This is done so that measurements of the same cloud volumes from parameterized versions of the 2DC and 2D-S can be compared with one another."

Line 44: Add Garrett et al.: Small, highly reflective ice crystals in low-latitude cirrus, GRL 2003.

Garrett et al. (2003) has been referenced and remarked on as follows. "Garrett et al. (2003) estimated that small ice crystals, with equivalent radii less than 30 microns, contributed in excess of 90% of total shortwave extinction during the NASA Cirrus Regional Study of Tropical Anvils and Cirrus Layers-Florida Area Cirrus Experiment

(CRYSTAL-FACE)."

Line 72: "which results jibes" is awkward–please rephrase.

Line 72 rephrased to "in agreement with Lawson (2011)".

Line 104-108: Perhaps a simple diagram would be helpful here to eludicate the method and steps used?

The text has been changed to the following, and a new Figure 1 has been inserted as shown below.

The comparison strategy, in short is as follows. The D05/D14 parameterizations consist of normalized, "universal" cirrus PSDs to which PSD moments are applied as inputs. The results of so doing are sets of parameterized 2DC PSDs—both shatter-corrected and uncorrected. To make the comparison, the same moments from 2D-S-measured PSDs are applied to the D05/D14 parameterizations in order to simulate what the shatter- and non-shatter-corrected 2DCs would have measured had they flown with the 2D-S. Then, a "universal" PSD derived from the 2D-S itself is computed in order to make a fair comparison. The moments from the 2D-S-measured PSDs are applied to the 2D-S "universal" PSD and it is then seen whether the older datasets differ statistically from the newer in their derived cirrus bulk properties. This procedure is illustrated in Fig. 1.

Line 164: Why not use the actual size distributions?

The idea had been to see how the $\sim$2nd moment of the fit PSD changed with varying degrees of truncation.

Line 166: Please quantify "nominally matches", particularly since the data aren't shown.

The word "nominally" has been replace with "qualitatively".

Line 339: Is it really the "true" value?
No, I suppose not really the true values. The wording is changed to indicate that "true" means derived directly from the measurements.

Line 369: Missing subscript in NT. Corrected.

Line 376: Delete this sentence as it's not really necessary?

Redundant sentence deleted.

Lines 387: A long and wordy sentence. Suggest breaking it up for clarity.

Long sentence split apart.

Line 391: If your other work giving better alternatives to the Gamma distribution is now published, please refer to it here.

Unfortunately, this is still in the submission stage and not yet published. Therefore, a reference to my dissertation is inserted here.

Line 396-398: Redundant with statements in prior paragraph; remove.

Redundant statements have been deleted.

Acknowledgements: No acknowledgements to those scientists who provided the field data?

Thank you for pointing out this oversight. It has been corrected by inclusion of the following text.

The author gratefully acknowledges the SPartICus, MACPEx, and TC4 science teams for the collection of data used in this study. TC4 and MACPEx data were obtained from the NASA ESPO archive, which may be accessed online at https://espoarchive.nasa.gov/archive/browse/. The SPartICus data were obtained from DOE ARM archive and may be accessed online at http://www.archive.arm.gov/discovery/#v/results/s/fiop::aaf2009Sparticus. In particular, the author acknowledges Dr. Paul Lawson and SPEC, Inc. for all 2D-S data

collected in the field, to Dr. Andrew Heymsfield for the PIP data used from TC4, and to Dr. Linnea Avallone for CLH data used from MACPEx.

Please also note the supplement to this comment:
http://www.atmos-meas-tech-discuss.net/amt-2017-48/amt-2017-48-AC1-supplement.pdf

————————————————

---

## Author Comment (AC2) · 29 Jun 2017

My response is included here in plain text. It is also attached as a pdf document, along with a marked up draft, for easier consideration.

xReview: Minor revisions I agree with the previous reviewers in general. This is a pretty clean cut topic for this paper which adds statistical consistency to a long standing problem in ice measurement. Further, looking at the references, this is clearly one

paper in a long series originating with the author's days at Utah and beyond. While the topic is clean cut, the paper is nevertheless difficult to follow at times, and the author could do much to improve readability, and hopefully in time, his h-index. Indeed, it is overly terse at times. One previous reviewer noted that the introduction could use a bit of background. I certainly concur with that. Even though this has been reviewed in several other papers, it is good for a paper to be complete. Not only to be tied in more completely with the previous literature base, but also with the author's current lien of thought. I would also say the final results and discussion could also be worked on. For example, the author states to the effect that old data is still usable, provided previously described caveats are respected. Actually going through the paper several times, it was not clear what all of these are. Even though the smaller ice sizes can be mitigated for the bulk moments, what does this mean for say a forward optical model? Perhaps a separate conclusions, or discussion and conclusions as distinct from results, be provided that provides a bulletined list of what are the key take away points-sort of a recipe card. I would also suggest that figure captions be more verbose spelling out variables when convenient. Similarly, laying out in a bulletining form or table the different instruments and processing would help. Other than these comments, my opinion matches those of the previous reviews: the paper oscillates between very formal writing, and conversational vernacular(e.g., jibes, right off the bat, etc); a diagram laying out the steps; . One point that requires emphasis as pointed out by reviewer 3 is the lack of data provider documentation in the acknowledgements. Indeed, by downloading data from the NASA servers not only did the author agree to acknowledge where the data came from, but actually offer coauthor ship to the data providers. Often for this sort of thing they will simply ask for acknowledgement, but the offer does need to be made. Be well.

Response to Referee J. Reid

I am grateful for your thoughtful review. I will attempt to address your remarks (in red) in order.

While the topic is clean cut, the paper is nevertheless difficult to follow at times, and the author could do much to improve readability... Indeed, it is overly terse at times.

This point is well taken. I made a number of changes to make notation coherent and consistent both within the text and within the figures, to make the tone more uniform, and to give better explanations. Rather than document all of those changes here, I'll simply put the marked up paper on with this reply.

One previous reviewer noted that the introduction could use a bit of background. I certainly concur with that. Even though this has been reviewed in several other papers, it is good for a paper to be complete. Not only to be tied in more completely with the previous literature base, but also with the author's current lien of thought.

I fleshed out the literature review quite a bit and tried to provide more context. In so doing, hopefully my own current lien of thought is better fleshed out. The following paragraphs were added to the introduction.

[revised manuscript text omitted]

. . . a diagram laying out the steps. . .

I put in improved text (below) about the steps and an accompanying figure (see the marked up draft).

The comparison strategy, in short is as follows. The D05/D14 parameterizations consist

of normalized, "universal" cirrus PSDs to which PSD moments are applied as inputs. The results of so doing are sets of parameterized 2DC PSDs—both shatter-corrected and uncorrected. To make the comparison, the same moments from 2D-S-measured PSDs are applied to the D05/D14 parameterizations in order to simulate what the shatter- and non-shatter-corrected 2DCs would have measured had they flown with the 2D-S. Then, a "universal" PSD derived from the 2D-S itself is computed in order to make a fair comparison. The moments from the 2D-S-measured PSDs are applied to the 2D-S "universal" PSD and it is then seen whether the older datasets differ statistically from the newer in their derived cirrus bulk properties. This procedure is illustrated in Fig. 1.

I would also say the final results and discussion could also be worked on. For example, the author states to the effect that old data is still usable, provided previously described caveats are respected. Actually going through the paper several times, it was not clear what all of these are. Even though the smaller ice sizes can be mitigated for the bulk moments, what does this mean for say a forward optical model? Perhaps a separate conclusions, or discussion and conclusions as distinct from results, be provided that provides a bulletined list of what are the key take away points-sort of a recipe card.

I've cleaned up the last section as well, summarizing the final points in a bulletined list as suggested. I'll refer you to the attached revision for a discussion on psd shape and optical models.

One point that requires emphasis as pointed out by reviewer 3 is the lack of data provider documentation in the acknowledgements. Indeed, by downloading data from the NASA servers not only did the author agree to acknowledge where the data came from, but actually offer coauthor ship to the data providers. Often for this sort of thing they will simply ask for acknowledgement, but the offer does need to be made.

You are quite correct about the acknowledgements. Co-authorship was offered to the data providers, which was declined. Thank you for pointing out my oversight in not

including the data sources in the acknowledgements. References to the data sources are also given.

Please also note the supplement to this comment: https://www.atmos-meas-tech-discuss.net/amt-2017-48/amt-2017-48-AC2-supplement.pdf

————————————————————————

---

## Author Comment (AC3) · 29 Jun 2017

My response is here in plain text, also attached as a pdf document for easier reading.

Response to Darrell Baumgardner

This study is a logical and complementary follow-up of the Delanoë et al. (2005,2014) evaluations that provided parameterizations of cirrus size distributions based on a large

set of measurements taken in both mid-latitude and tropical environments. The author has provided a detailed analysis using more recent measurements with a more modern imaging probe to address an important question: "Given what we now know about the impact of crystal shattering on measurements by cloud particle spectrometers, can historical data sets be trusted"? I think that this study has answered that question, at least with respect to cirrus clouds. In addition, even though the instrument that is used in this study has a faster response time than the earlier 2D-C and 2D-P, and marginally larger sample volume, the results of the current study would suggest that such instrument improvements really have minor impact on the overall statistical robustness of the previous measurements and may also only be marginally more accurate, especially given the many other uncertainties that the new instrument has not overcome. In particular, there remain major uncertainties due to unknown ice density and shape n the third dimension that lead to large error bars in derived bulk parameters. It is only at the very smallest sizes where there is a clear difference between current and previous measurements; however, even when there are several orders of magnitude difference in concentration at these sizes, the propagated error in effective radius, IWC and reflectivity is surprisingly small. What I think would be a useful, and perhaps even necessary, addition to this paper would be to include in Figs. 7&8 the relative errors and standard deviations that are reported in Delanoë et al. (2005,2014) where they compare their data sets against the parameterization. That would then put into context the current comparison with the parameterizations with the original, hence bringing closure. The other very important source of uncertainty that the author side steps is that of oversizing of out-of-focus ice crystals (Korolev, 2007). Although a correction for this issue has not yet been provided, such as has been done for water droplets, measurements in cirrus clearly show crystal images that are out of focus and that should be sizecorrected. These might even be the source of the "bump" in the size distributions, i.e. a certain fraction of the particles in that size interval most certainly are smaller crystals out of focus. This bump is also seen in the Delanoë et al. (2005,2014) studies; however, whereas the bump occurs in the current study at a Deq/DM <1, in Delanoë et

al. (2005,2014) the bump is right at 1. How does the author explain this? Lastly, the author refers to three of his papers that have not yet been published. These references should be removed since, as a reviewer, I was unable to access them.

I thank you for your time in providing a thoughtful review. I will attempt to address your remarks (in red) in order.

What I think would be a useful, and perhaps even necessary, addition to this paper would be to include in Figs. 7&8 the relative errors and standard deviations that are reported in Delanoë et al. (2005,2014) where they compare their data sets against the parameterization.

I must confess that I entirely misread this comment at first and added error bars to show standard error in the means and standard deviations to those figures. However, now that I have overcome my stupor of thought and understand your comment correctly, I'm not sure that I can read the numbers off those charts accurately enough to replot them.

The other very important source of uncertainty that the author side steps is that of oversizing of out-of-focus ice crystals (Korolev, 2007). Although a correction for this issue has not yet been provided, such as has been done for water droplets, measurements in cirrus clearly show crystal images that are out of focus and that should be sizecorrected. These might even be the source of the "bump" in the size distributions, i.e. a certain fraction of the particles in that size interval most certainly are smaller crystals out of focus. This bump is also seen in the Delanoë et al. (2005,2014) studies; however, whereas the bump occurs in the current study at a Deq/DM <1, in Delanoë et al. (2005,2014) the bump is right at 1. How does the author explain this?

I have remarked on the out-of-focus problem in the revamped introduction. However, I have no good explanation for the shifting of the bump. I decided to leave that unaddressed rather than risk proffering a bad explanation. The additional text in the introduction follows.

[revised manuscript text omitted]

Lastly, the author refers to three of his papers that have not yet been published. These references should be removed since, as a reviewer, I was unable to access them.

I removed them and replaced them with a simple reference to my dissertation.

Please also note the supplement to this comment: https://www.atmos-meas-tech-discuss.net/amt-2017-48/amt-2017-48-AC3-supplement.pdf

---

## Author Comment (AC4) · 29 Jun 2017

I have the response here in plain text, but the attached pdf will be much easier to read.

The material in this manuscript is suitable for publication in amt. It gives a useful comparison between an older particle probe, the 2DC, and the newer probe, the 2DS, thought to provide more accurate ice crystal information. A compilation of the parameterization and normalization of many ice crystal size distributions measured by both

probe types is used in an attempt to adjust the older probe data to make that data more reliable.

1. The paper needs a careful review concerning the lack of definition of some given variables. For example, what is a_mi and b_mi in Eq. (4), what is D_eq in the Figures, what is subscript I ?

2. The accuracies of the density/dimension and mass/dimension relationships used in the paper are not discussed, even though they may affect the conclusions reached. A comment on such a possible affect.

3. The data for D05/D014 is listed as starting at 25 um; whereas the data for the 2DS starts at 15 um. Is this taken into account in the comparisons?

4. The author points out the difficulty of the probes measuring the smallest ice crystals, given that the probes can create errors due to uncorrected crystal shattering and other reasons. His sentence associated with small crystals (line 181) "It is therefore felt that the averaging approach is justified" is inconsistent with this difficulty.

5. The paper only deals with integrated ice-crystal properties, but it also points out that the nature of the ice-crystal size distribution should also play a significant role in probe performance. The latter is not dealt with in the paper. It would be helpful for the author to comment on what might be done to improve the size information on the smallest ice crystals that can dominate under certain atmospheric conditions (e.g., Heymsfield et al., 2010, JAS, 67, 3303-3318). For example, can forward scattering probes that respond to small particles be used for ice crystal measurements (e.g., Gerber and DeMott, 2014, JTECH, 31, 2145-2155)?

6. The impressive Appendix is not essential for the conclusions reached in the paper. Deletion of the Appendix is recommended.

Response to Anonymous Referee #2

[Figure]

Thank you for your thoughtful and helpful review. I will address your remarks in order.

1. This point is well taken. I have gone over the text and have removed inconsistencies (viz., that Deq and De are supposed to be the same), have explicitly described each variable and its subscript, and have removed the error of always using the letter "i" for every subscript. Rather than document each change here, I've attached the marked-up manuscript to this reply.

2. In fact, there is an unfortunately high amount of uncertainty in these relations. It was felt that the best that could be done was to use the same relations in this paper as in D05/D14 so as to keep that part of the comparison consistent. This, of course, assumes the same overall mix of particle habits was encountered between the PSD datasets. This is now noted in the discussion section.

3. No, it is not. In light of the difference found, that is well worth pointing out and is done so in the final section.

4. I think perhaps that I've not worded that sentence well and that it is redundant. The "averaging approach" is adopted for smoothing out Poisson counting noise, not for ameliorating measurement problems such as shattering. The shattered particle removal post-processing (performed by the instrument team) is aimed at that. The sentence in question has been removed, and the following sentence has been inserted at line 157 (given with the sentence prior for context).

"In the first exercise, fifteen-second temporal averages were performed along with truncating zero through two of the smallest size bins while only the unimodal fits (chosen according to a maximum likelihood ratio test [Wilks, 2006]) were kept. This exercise was performed first so as to prevent the most spurious size bins' interfering with the smoothing out of Poisson counting noise."

5. This matter is now dealt with in the final section.

1) Finally, it is important to note that this study does not specifically consider PSD

shape. (For a more detailed discussion on cirrus PSD shape and on the efficacy of the gamma distribution, please refer to Schwartz [2014].) This is a critical component of the answers to Korolov et al.'s (2013b) original two questions. Mitchell et al. (2011) demonstrated that for a given effective diameter and IWC, the optical properties of a PSD are sensitive to its shape. Therefore, PSD bimodality and concentrations of small ice crystals are critical to realistically parameterizing, cirrus PSDs, to modeling their radiative properties and sedimentation velocities, and to mathematical forward models designed to infer cirrus PSDs from remote sensing observations (Lawson et al., 2010; Mitchell et al, 2011; Lawson, 2011). In order to improve knowledge on PSD shape, as well as to develop statistical algorithms for correcting historical PSD datasets so that PSD shapes are corrected along with computations of bulk properties, it will be necessary to make use of instruments that can provide reliable measurements of small ice crystals beneath the size floors of both the 2DC and the 2D-S. Recent studies such as Gerber and DeMott (2014) have provided aspherical correction factors for particle volumes and effective diameters measured by the FSSP. However, the author expects that this problem will ultimately be resolved by the continued technological development of new probes such as the HOLODEC.

6. The Appendix has been removed.

Please also note the supplement to this comment:
https://www.atmos-meas-tech-discuss.net/amt-2017-48/amt-2017-48-AC4-
supplement.pdf
* * *